# Evolution from unimolecular to colloidal-quantum-dot-like character in chlorine or zinc incorporated InP magic size clusters

Yongju Kwon[1,4], Juwon Oh[2,3,4], Eunjae Lee[1], Sang Hyeon Lee[2], Anastasia Agnes[1], Gyuhyun Bang[1], Jeongmin Kim[1], Dongho Kim[2✉] & Sungjee Kim [1✉]

Magic-sized clusters (MSCs) can be isolated as intermediates in quantum dot (QD) synthesis, and they provide pivotal clues in understanding QD growth mechanisms. We report syntheses for two families of heterogeneous-atom-incorporated InP MSCs that have chlorine or zinc atoms. All the MSCs could be directly synthesized from conventional molecular precursors. Alternatively, each series of MSCs could be prepared by sequential conversions. 386-InP MSCs could be converted to F360-InP:Cl MSCs, then to F399-InP:Cl MSCs. Similarly, F360-InP:Zn MSCs could be converted to F408-InP:Zn MSCs, then to F393-InP:Zn MSCs. As the conversion proceeded, evolution from uni-molecule-like to QD-like characters was observed. Early stage MSCs showed active inter-state conversions in the excited states, which is characteristics of small molecules. Later stage MSCs exhibited narrow photoinduced absorptions at lower-energy region like QDs. The crystal structure also gradually evolved from polytwistane to more zinc-blende.

[1] Department of Chemistry, Pohang University of Science and Technology, Pohang 37673, South Korea. [2] Department of Chemistry and Spectroscopy Laboratory for Functional π-Electronic Systems, Yonsei University, Seoul 03722, South Korea. [3] Department of Chemistry, University of California, Berkeley, CA 94720, USA. [4]These authors contributed equally: Y. Kwon, J. Oh. ✉email: dongho@yonsei.ac.kr; sungjee@postech.ac.kr

Colloidal semiconducting quantum dots (QDs) have unique optical, physical, and chemical properties, which have led to innovative applications such as imaging probes, photodetectors, photovoltaics, and light-emitting devices[1–3]. InP QDs have shown commercial viability for display applications, and have accelerated the necessity for high-quality InP QD synthesis[4,5]. To rationally design high-quality InP QDs ultimately in molecular precision, the growth mechanisms from molecular precursors to QDs must be better understood. In this regard, there have been intensive studies, which revealed that InP QDs appear to be quite disparate from Cd-chalcogenide QDs in their growth mechanism[6–8].

Mechanistic studies on QDs often identify magic-sized clusters (MSCs) as intermediates between molecular precursors and final QDs[7,9,10]. InP MSCs have been reported as intermediates in the formation of QDs[6]. MSCs are distinguished from a small size of nanoparticles (NPs), because MSCs are thermodynamically more stable than expected ones in similar size NPs. A colloidal semiconductor MSC usually consists of a small number (typically less than hundreds) of atoms in a diameter <3 nm. Absorption features persistently occurring at fixed wavelengths are the characteristics constantly observed for MSCs, and MSCs are typically categorized by the wavelengths and widths of such features for the lowest energy electronic transition (LEET)[11,12].

MSCs are composed of a specific number of atoms and possess a high degree of uniformity, although the required degree of uniformity is not unanimously defined. Cossairt and co-workers published a seminal paper reporting synthesis of an InP MSC from molecular precursors of indium acetate ($In(Ac)_3$), myristic acid (HMA), and tris(trimethylsilyl)phosphine (($TMS)_3P$), which exhibits the dominant absorption peak at 386 nm (denoted as 386-MA InP MSC)[6]. The structure has been clearly elucidated by single crystal X-ray crystallography as $In_{37}P_{20}(O_2CR)_{51}$ with the definitive inorganic structure and ligand configuration[13]. The structural uniformity of 386-MA InP MSC is monodisperse. However, some other MSCs are suspected to be disordered and have structural information not clearly analyzed[14–16], possibly as a result of isomerism in both the inorganic crystal structures and surface-binding ligand configurations. When the monodispersity of an MSC is not confirmed and disorders are suspected, MSCs that show a persistent peak are denoted herein as a family of MSCs, abbreviated as F(persistent wavelength)(components)-MSCs. For example, InP MSCs that show a persistent peak at 397 nm and that bear octadecylphosphonic acids (ODPAs) on their surface are denoted as F397-ODPA InP MSCs[6].

InP MSCs of different sizes or structures, and InP MSCs of different surface ligands have been reported[6,13,17]. However, InP MSCs that host heterogeneous atoms have not been studied in detail. Cation exchange of InP MSCs with Cd ions has been reported; however, the work mainly focused on the structural transformation to $Cd_3P_2$ clusters[18]. Here, we report two InP MSC families: one that incorporates chlorine (Cl) atoms (InP:Cl MSCs), and one that incorporates zinc (Zn) atoms (InP:Zn MSCs). Halides can strongly bind to QD facets, and binding of Cls to particular facets may stabilize QDs[19]. For example, InAs and PbS QDs post-treated by Cl showed higher photovoltaic properties owing to the Cl-terminated stabilized surfaces[20,21]. Such facet-specificity induced by Cl bindings can be also useful to direct the growth InP QDs and to control their shape[19]. Zn is also important for InP QDs because bright InP-based QDs typically have Zn-chalcogenide shell layers (e.g., ZnSe, ZnS). Alloy formation at core-shell interfaces (InZnSe; InZnS), and inter-diffusion of Zn into InP cores strongly affect the emission properties of QDs[22–24].

Here, we report two families of Cl-incorporated InP MSCs (F360-InP:Cl MSCs, F399-InP:Cl MSCs) and three families of Zn-incorporated InP MSCs (F360-InP:Zn MSCs, F408-InP:Zn MSCs,

F393-InP:Zn MSCs) (Fig. 1). All of these MSCs could be directly synthesized from molecular precursors. Alternatively, 386-InP MSCs were used as the starting material for the conversion into F360-InP: Cl and F399-InP:Cl MSCs (Fig. 1a). Interestingly, 386-InP and F360-InP:Cl MSCs differ from F399-InP:Cl MSCs in their main structures, which leads to that the former two MSCs have molecular-like characters and F399-InP:Cl MSCs show QD-like characters. Similarly, F360-InP:Zn MSCs could be converted to F408-InP:Zn MSCs, which could be further converted to F393-InP: Zn MSCs (Fig. 1b). The transition from F408-InP:Zn MSCs to F393-InP:Zn MSCs was identified as a ligand change reaction. Analogous to the InP:Cl MSCs, F360-InP:Zn MSCs and F408-InP:Zn MSCs/ F393-InP:Zn MSCs show a molecular- and QD-like character, respectively, upon their structures and the Zn incorporation in InP MSCs leads to Zn-mediated trap states. For both InP:Cl MSCs and InP:Zn MSCs, evolutions from uni-molecule-like or small-molecule-like characters gradually to QD-like behaviors were observed.

## Results

**Syntheses of F360-InP:Cl MSCs and F399-InP:Cl MSCs.** As a representative direct synthesis of F360-InP:Cl MSCs and F399-InP:Cl MSCs, $In(Ac)_3$, HMA and indium chloride ($InCl_3$) were mixed in 1-octadecene (ODE). The molar mixing ratio was 1:3:0.6 for F360-InP:Cl MSCs and 1:3:1 for F399-InP:Cl MSCs. ($TMS)_3P$ dispersed in ODE was injected into the In mixture at 50 °C for F360-InP:Cl MSCs and 80 °C for F399-InP:Cl MSCs. Immediately after injection, the absorption spectra was featureless to ~450 nm (Fig. 2a, b). Such featureless absorption may be a signal of non-crystalline InP clusters that do not have a specific structure. As the reaction proceeded, the featureless absorption profile subsided and a persistent absorption peak emerged and dominated; this peak was at 360 nm for F360-InP:Cl MSCs and at 399 nm for F399-InP:Cl MSCs.

The fate of F360-InP:Cl MSCs and F399-InP:Cl MSCs was more dependent on the reaction temperature $T_R$ than the mixing ratio of In precursor. Control experiments were performed using the precursor ratio optimal for F360-InP:Cl MSCs (1:3:0.6) but at $T_R = 110$ °C or 80 °C instead of 50 °C; F399-InP:Cl MSCs were produced at a rate significantly slower than the representative F399-InP:Cl MSCs synthesis at both elevated $T_R$ (Supplementary Fig. 1a, b). At both precursor ratios used for F360-InP:Cl MSCs and F399-InP:Cl MSCs, no reaction occurred at room temperature (RT), and yielded uncontrolled growth into InP NPs at $T_R >$ 150 °C (Supplementary Fig. 1c, d).

F360-InP:Cl MSCs requires less $InCl_3$ than F399-InP:Cl MSCs. To further investigate the $InCl_3$ effect, the $InCl_3$ ratio was greatly reduced to 1:3:0.08 at 80 °C; the extreme reduction of $InCl_3$ supply drove the reaction to F360-InP:Cl MSCs (Supplementary Fig. 1e), whereas both of the representative $InCl_3$ molar ratios (1:3:1 and 1:3:0.6) at 80 °C yielded F399-InP:Cl MSCs. Excessive $InCl_3$ supply typically yielded bulk precipitates (Supplementary Fig. 1f).

To better understand the growth mechanisms of F360-InP:Cl MSCs and F399-InP:Cl MSCs, conversion syntheses of those MSCs were developed from 386-InP MSCs. 386-InP MSCs have the polytwistane crystal structure with pseudo-$C_2$ symmetry[13,25]. Because of this well-characterized structure, 386-InP MSCs were used as a starting species for conversion syntheses to MSCs that incorporate Cl. 386-MA InP MSC was synthesized using a reported protocol[6]. As-synthesized 386-MA InP MSCs were further reacted with 56 equivalents of $InCl_3$, 1.1 times MA ligands per MSC. Aliquots of a fixed small amount were taken over time to follow the absorption (Fig. 2c). Upon addition of $InCl_3$, the absorption continuously red-shifted. Within 1 h, the peak at 386 nm disappeared and two new peaks began to be resolved: one peak at 416 nm that appeared first and led the initial red-shift, and the

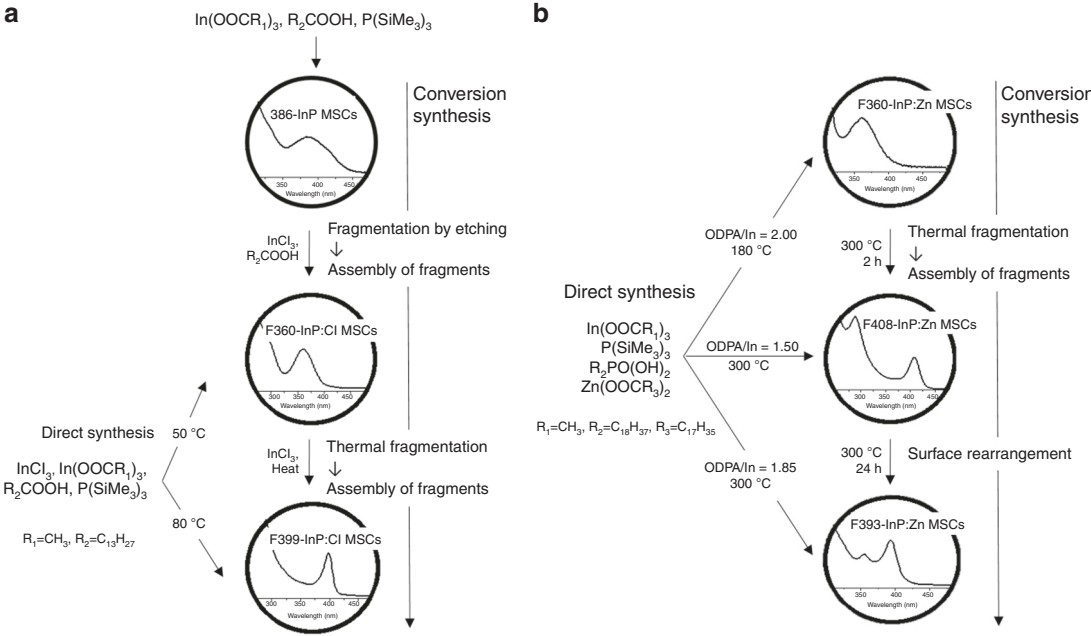

**Fig. 1 Schematic illustration.** Direct or conversion synthesis pathway for **a** Cl-incorporated InP MSCs and **b** Zn-incorporated InP MSCs.

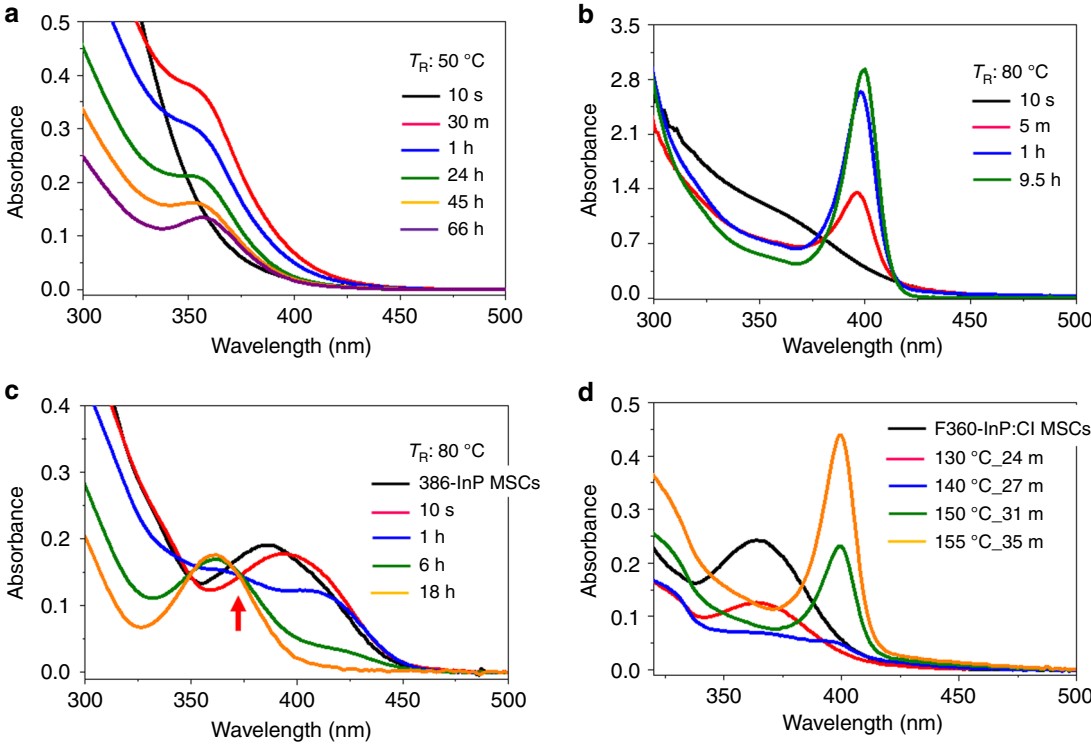

**Fig. 2 Syntheses of F360-InP:Cl MSCs and F399-InP:Cl MSCs.** UV–vis absorption spectra of aliquots taken over time during **a** direct synthesis of F360-InP:Cl MSCs from In(Ac)₃, HMA, InCl₃, and (TMS)₃P at 50 °C. The molar ratio of In(Ac)₃, HMA, to InCl₃ is 1:3:0.6. **b** Direct synthesis of F399-InP:Cl MSCs from In(Ac)₃, InCl₃, HMA, and (TMS)₃P at 80 °C. The molar ratio of In(Ac)₃, HMA to InCl₃ is 1:3:1. **c** Conversion synthesis of F360-InP:Cl MSCs from as-synthesized 386-MA InP MSCs using 56 equivalents of InCl₃ at 80 °C. (Red arrow: isosbestic point) **d** Conversion synthesis of F399-InP:Cl MSCs from F360-InP:Cl MSCs by heating.

other peak at 360 nm to which the sample was eventually merged. 386-InP MSCs were converted into F360-InP:Cl MSCs via a 416 nm intermediate species (416-IS) that showed the LEET at 416 nm. 416-IS peak was fitted by a Gaussian peak that has 20 nm half width at half maximum (HWHM) (Supplementary Fig. 2). Though

416-IS could not be purely isolated, a mixture sample rich in 416-IS could be obtained during a conversion from 386-InP MSC to F360-InP:Cl MSC at a lowered temperature. Combined analysis based on the absorption, X-ray diffraction (XRD), and elemental analysis suggests 416-IS to be another InP:Cl MSCs

(Supplementary Fig. 3). The added $InCl_3$ in situ generates HCl and etches 386-MA InP MSCs as evolving $PH_3$ (vide infra). Composition analysis showed more In-rich for 416-IS than 386-MA InP MSC. We speculate that 416-IS has the inorganic skeleton that is from partially etched 386-MA InP MSC. Similarly to this, 386-MA InP MSCs have been reported to be etched by amines and to show the red-shifted absorption[26]. After 18 h at 80 °C, 416-IS disappeared completely and only F360-InP:Cl MSCs survived to show a persistent peak at 360 nm with HWHM = 20 nm. At 80 °C, a seemingly isosbestic point (Fig. 2c, arrow) was observed in conversion from 416-IS to F360-InP:Cl MSCs. However, this point was not observed when the conversion was slowed by conducting the reaction at 50 °C or 25 °C (Supplementary Fig. 4b, c); this absence suggests that the conversion is not one-to-one transformation but involves other intermediates that have not been properly detected. We speculate that 416-IS fragmented to form parts that assembled into F360-InP:Cl MSCs. The conversion from 416-IS to F360-InP:Cl MSCs accompanies rise and fall in broad UV absorption (Supplementary Fig. 4d). The broad UV absorption is attributed to the fragments from 416-IS. When the conversion $T_R$ was lowered to 50 °C or to 25 °C, the appearance and disappearance of broad UV absorption became more notable (Supplementary Fig. 4e, f).) Different $T_R$ may have affected differently for the kinetics of fragmentations and their assemblies[27]. Different $T_R$ may have also yielded different fragments and partial assemblies of which resulted in the same final product. Alternatively, 416-IS may have gone through some sort of amorphonization before crystallize to F360-InP:Cl MSCs.

We explored whether F360-InP:Cl MSCs that had been prepared by conversion from 386-MA InP MSCs could be further converted to another species. Without purification, as-synthesized F360-InP:Cl MSCs were slowly heated to 155 °C and the absorption was monitored (Fig. 2d). The solution became transparent as the peak at 360 nm rapidly decreased at $T_R > 120$ °C, which was subsequently followed by an emergence of strong yellow tint in the solution with a new sharp peak at 399 nm (HWHM = 8 nm) at $T_R > 140$ °C. Upon heating, F360-InP:Cl MSCs were converted into F399-InP:Cl MSCs. The conversion is considered to proceed by thermal fragmentation and subsequent reassemblies and rearrangements. The appearance and disappearance of broad UV absorption were observed during the conversion from F360-InP:Cl MSCs to F399-InP:Cl MSCs (Supplementary Fig. 5), as was observed during production of F360-InP:Cl MSCs from 386-InP MSCs via 416-IS.

When the conversion reaction conditions from 386-InP MSCs to F360-InP:Cl MSCs or to F399-InP:Cl MSCs were varied, they returned patterns similar to those observed for direct syntheses of F360-InP:Cl MSCs and F399-InP:Cl MSCs. Starting from 386-MA InP MSCs at $T_R < 80$ °C, conversion was halted at F360-InP:Cl MSCs and did not further proceed to F399-InP:Cl MSCs (Supplementary Fig. 4). At $T_R > 110$ °C conversion to F399-InP:Cl MSCs occurred and at $T_R > 180$ °C such conversion was rapidly followed by further growth to NPs (Supplementary Fig. 6). Heat-up experiment to 200 °C also yielded similar results with zinc-blende a-few-nm InP NPs (Supplementary Fig. 7). When the amount of $InCl_3$ was reduced (20 equivalents instead of 56 equivalents), even the conversion to F360-InP:Cl MSCs was incomplete (Supplementary Fig. 8a). Reduction of $InCl_3$ amount only to 40 equivalents resulted in a complete conversion to F360-InP:Cl MSCs, but the conversion did not proceed to the conversion to F399-InP:Cl MSCs (Supplementary Fig. 8b). This result indicates that extra $InCl_3$ is needed for the subsequent conversion from F360-InP:Cl MSCs to F399-InP:Cl MSCs. We hypothesize that the thermally fragmented intermediates from F360-InP:Cl MSCs must react with $InCl_3$ before they can properly assemble and rearrange into F399-InP:Cl MSCs. Excess use of $InCl_3$ greatly accelerated the conversion from

386-MA InP MSCs to F360-InP:Cl MSCs and the conversion could be achieved at RT (Supplementary Fig. 8e, f). However, the conversion from F360-InP:Cl MSCs to F399-InP:Cl MSCs needed thermal activation even with excess $InCl_3$; the observation corroborates our hypothesis of thermal fragmentation from F360-InP:Cl MSCs to F399-InP:Cl MSCs. Nevertheless, the presence of excess $InCl_3$ could lower the conversion $T_R$ to 110 °C (Supplementary Fig. 9a, b).

**Studies on the mechanism of conversion from 386-InP MSC to F360-InP:Cl MSCs.** To further understand the conversion mechanism from 386-InP MSCs to F360-InP:Cl MSCs, the effect of free carboxylic acids was studied. Oleate (OA)-capped InP MSCs that showed a prominent absorption peak at 386 nm (386-OA InP MSCs), were prepared (Supplementary Methods). 386-OA InP MSCs undergo conversion to F360-InP:Cl MSCs and to F399-InP:Cl MSCs as 386-MA InP MSCs do. Oleic acid (HOA) is more soluble than HMA in organic solvents at RT. HOA is also better suited than HMA for $^1H$ NMR studies. Conventional aliphatic carboxylic acids such as HMA are not optimal for quantitative $^1H$ NMR analysis because the spectra have large overlapping and polydispersive peaks. In contrast, alkene-containing ligands such as HOA enable simple spectral discrimination between free and bound species by exploiting the spectral window at 4.0–6.0 ppm.

As-synthesized 386-OA InP MSCs and purified 386-OA InP MSCs were, respectively, used to convert to F360-InP:Cl MSCs. To remove remaining small molecules such as free HOA, the products were purified in a nitrogen-filled glove box by gel-permeation chromatography (GPC) using styrene divinylbenzene beads with 1% cross-linkage. The removal of free HOA was confirmed by $^1H$ NMR (Supplementary Fig. 10). Upon addition of $InCl_3$ (56 equivalents to MSC) and heating to 130 °C, as-synthesized 386-OA InP MSCs were rapidly converted to F360-InP:Cl MSCs, whereas purified 386-OA InP MSCs remained unconverted and suddenly grew to form InP NPs (Fig. 3a, b). The as-synthesized 386-OA InP MSCs had free HOA, which was absent for the purified 386-OA InP MSCs. Free HOA should have a pivotal role in the conversion to F360-InP:Cl MSCs. To test this hypothesis, the purified 386-OA InP MSCs were heated with 56 equivalents of $InCl_3$ and 112 equivalents of HOA; as a result, they were converted to F360-InP:Cl MSCs (Fig. 3c).

Proton-coupled $^{31}P\{^1H\}$ NMR spectra were taken for 386-OA InP MSCs and for F360-InP:Cl MSCs (Fig. 3d). 386-InP MSCs showed 10 phosphorus peaks in the chemical shift range between −190 and −250 ppm; this result matches well with the spectrum previously reported by others[25]. As 386-OA InP MSCs were converted to F360-InP:Cl MSCs, the 10 peaks merged into a broad peak at −220 ppm and concurrently a new quartet peak arose at −243 ppm. This new peak is assigned to $PH_3$ generated in situ. $InCl_3$ must have reacted with HOA and generated HCl in situ. Subsequently, the HCl may have etched 386-InP MSCs by removing P atoms. Leaching out P atoms as $PH_3$ may have been followed by fragmentation of the MSCs. The conversion from 386-InP MSCs to F360-InP:Cl MSCs may involve:

$$InCl_3 + 3RCOOH \rightarrow In(RCOO)_3 + 3HCl \qquad (1)$$

$$In_{37}P_{20}(RCOO)_{51} + 3x\,HCl \rightarrow In_{37-y}P_{20-x}(RCOO)_{51-3y}$$
$$Cl_{3x} + y\,In(RCOO)_3 + xPH_{3(g)}$$
$$R = C_{13}H_{27} \text{ or } C_{17}H_{33}$$
$$(2)$$

Using the elemental analysis data of the reactant and product (Table 1), the $x$ value herein can be assigned approximately as six.

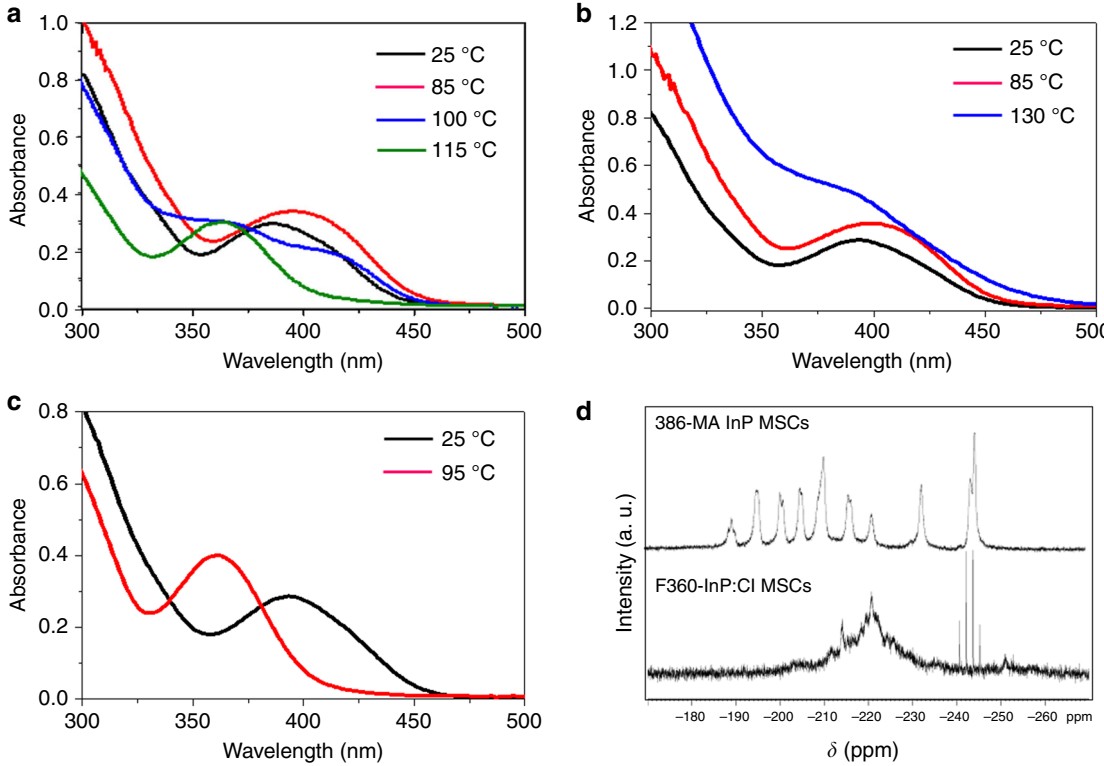

**Fig. 3 Conversion from 386-InP MSC to F360-InP:Cl MSCs.** UV−vis absorption spectra of aliquots taken over time **a** during conversion from as-synthesized 386-OA InP MSCs to F360 InP:Cl MSCs using 56 equivalents of $InCl_3$, **b** during conversion reaction from purified 386-OA InP MSCs using 56 equivalents of $InCl_3$, and **c** during conversion from purified 386-OA InP MSCs to F360 InP:Cl MSCs using 56 equivalents of $InCl_3$ and 112 equivalents of OA. **d** $^{31}P\{^1H\}$ NMR spectra of 386-OA InP MSCs and the converted F360-InP:Cl MSCs.

**Table 1 Chemical compositions (atomic%) of 386-InP MSCs, F360-InP:Cl MSCs, and F399-InP:Cl MSCs.**

| MSC | In[a] | P[a] | Cl[a] | MA[b] |
|---|---|---|---|---|
| 386-InP | 1.0 | 0.54 | – | 1.4 |
| F360-InP:Cl | 1.0 | 0.32 | 0.38 | 1.7 |
| F399-InP:Cl | 1.0 | 0.48 | 0.45 | 1.1 |

[a]Relative amount of In, P, and Cl was measured from EDX, and [b]amount of MA was deduced for the conditions satisfying the MSC charge neutrality.

However, the reactions stated above do not entirely describe the conversion reaction from 386-InP MSCs to F360-InP:Cl MSCs. When $y = 0$, the reaction describes an anionic exchange reaction between P and Cl. However, such case is unlikely because all the surface atoms are In for 386-InP MSCs.

Another control experiment was performed by reacting purified 386-OA InP MSCs with aqueous HCl instead of HOA and $InCl_3$, This experiment process also yielded F360-InP:Cl MSCs (Supplementary Fig. 11).

During the initial stage of conversion from 386-InP MSCs to F360-InP:Cl MSCs when the 416-IS appears, $^{31}P\{^1H\}$ NMR confirmed the existence of $PH_3$ (Supplementary Fig. 12); this result indicates that the 416-IS may be a partially etched species from 386-InP MSCs. HCl generation in situ is importantly involved in the multi-step conversion from 386-InP MSCs to F360-InP:Cl MSCs.

An NMR experiment was devised using chlorotrimethylsilane (TMS-Cl) as a Cl source instead of $InCl_3$. TMS-Cl can readily react with carboxylic acids and produce HCl by esterification. It is a liquid that is well-soluble in most organic solvents, and fine control over the amount can be achieved. TMS-Cl toluene solution was added sequentially to as-synthesized 386-OA InP MSCs (Fig. 4a) in toluene (13.5 mM) to total TMS-Cl amounts of 0.6, 1, 5, 15, 50, and 100 equivalents per MSC, and absorption and $^{31}P\{^1H\}$ NMR were simultaneously monitored (Fig. 4b, c). In $^{31}P$ NMR, three peaks at −201 ppm (integration corresponding to two P), −205 ppm (two P), and −210 ppm (four P) are assigned for eight phosphorus atoms of P5, P6, P9, P13, P14, P15, P19, and P20 (Fig. 4a). 386-InP MSCs show $C_2$ symmetry, so P5, P6, P9, and P13 are equivalent to P20, P15, P14, and P19. Addition of 0.6 equivalents of TMS-Cl to 386-OA InP MSCs resulted in reduction of the peaks at −201 ppm and −205 ppm (Fig. 4c, dotted red box), which indicates change in the local environment at the two phosphorous positions. Concurrently, two new peaks emerged at −212 ppm and −237 ppm (Fig. 4c, black arrows). Such change suggests that selective replacements of OA ligands by Cl occurred at particular positions. As the amount of added TMS-Cl increased, the P10 peak at −192 ppm (Fig. 4c, red arrow) decreased and another new peak at −185 ppm (Fig. 4c, blue arrow) grew; this change may be a result of etching of In atoms that neighbor P10. The In atoms bonded to P10 have been reported to be quite acidic, to be readily etched by amines and to cause downfield shift in $^{31}P$ NMR[26]. Continued addition of TMS-Cl caused gradual broadening and merging of NMR peaks, and increasing growth of the $PH_3$ quartet peak was observed. The broadening in NMR peaks may originate from inhomogeneity and isomerism in F360-InP:Cl MSCs and also from dynamic change in the configuration of mixed ligands of Cl and OA.

**Characterizations of F360-InP:Cl MSCs and F399-InP:Cl MSCs.** $^{31}P\{^1H\}$ NMR spectra were obtained for 386-MA InP MSCs and InP:Cl MSCs in toluene-d8 after $PH_3$ removal by

## a  386-InP MSC

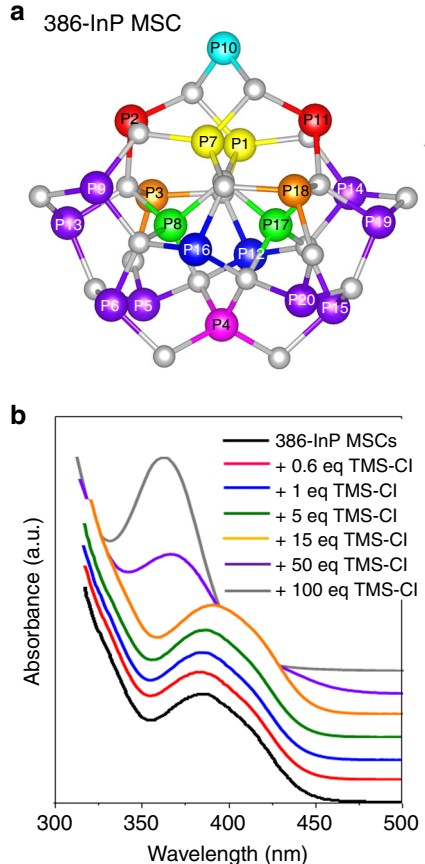

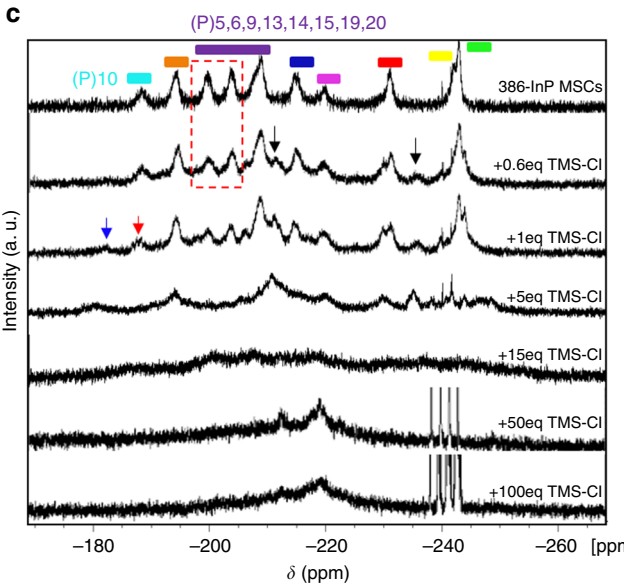

**Fig. 4 Early-stage conversion from 386-InP MSC to F360-InP:Cl MSCs. a** The view of $In_{21}P_{20}^{3+}$ core structure in 386-InP MSC. The structural figure of 386-InP MSC was drawn using reported crystallographic information file[13]. **b** Evolution of UV–vis absorption spectra of 386-OA InP MSCs with addition of TMS-Cl of 0.6, 1, 5, 15, 50, and 100 equivalents, relative to 386-InP MSC in toluene-d8. **c** Evolution in $^{31}P\{^1H\}$ NMR spectra of 386-OA InP MSCs upon the TMS-Cl addition; colored bars at top: represent corresponding phosphorus atoms of the same color in **a**. (Dotted red box: reduction of the peaks, black arrows: peaks emerging at +0.6 eq. addition, red arrow: P10 peak, blue arrow: peak emerging at +1 eq. addition).

degassing (Fig. 5a). InP:Cl MSCs were prepared by conversion using 386-MA InP MSCs. F360-InP:Cl MSCs capped by MA showed a broad phosphorus peak at −220 ppm (Fig. 5a). F360-InP:Cl MSCs may be partially etched from 386-InP MSCs with some additional rearrangements and incorporation of Cl. F360-InP:Cl might have a similar structure as the $In_{29}P_{14}$ species reported by others[25]. Broadening in F360-InP:Cl MSCs is attributed to inhomogeneous distributions and dynamics of two surface-binding molecules of MA and Cl. F399-InP:Cl MSCs exhibited an extremely broad peak at down-fielded −160 ppm with a hump at −230 ppm. The discrepancy in NMR resonance peak chemical shifts between F360-InP:Cl MSCs and F399-InP:Cl MSCs is presumably owing to the different crystal structures, as will be discussed later. Contrary to the very broad $^{31}P$ NMR peak, F399-InP:Cl MSCs showed a very narrow (HWHM = 8 nm) LEET absorption peak. F399-InP:Cl MSCs are suspected to be larger than F360-InP:Cl MSCs by an amount enough to develop inner crystal periodicity, which may increase the degeneracy as expressed by the narrow electronic transition, but the environmental discrepancy between the inner P atoms and outer near-surface atoms results in broad NMR peaks. The sizes were measured by transmission electron microscopy (TEM) as 1.7 nm, 1.7 nm, and 2.1 nm, respectively, for 386-MA InP MSCs, F360-InP:Cl MSCs, and F399-InP:Cl MSCs (Fig. 5b).

The In 3d X-ray photon spectroscopy (XPS) spectra for 386-MA InP MSCs, F360-InP:Cl MSCs, and F399-InP:Cl MSCs shows In $3d_{5/2}$ peaks at 444.69, 445.25, and 445.46 eV (Fig. 5c). Compared with 386-MA InP MSCs, the binding energies increased by 0.56 eV for F360-InP:Cl MSCs and by 0.77 eV for F399-InP:Cl MSCs. The large increase in the binding energies is indicative of the direct bindings of Cl to In atoms. More $InCl_3$ is required for the synthesis of F399-InP:Cl MSCs than for F360-InP:Cl MSCs. F399-InP:Cl MSCs should have more Cl on the surface than F360-InP:Cl MSCs, which resulted in the increased binding energy. Bulk $InCl_3$ shows a In $3d_{5/2}$ XPS peak at ~446 eV[28].

Energy dispersive X-ray (EDX) spectroscopy was performed to quantify the compositions of 386-MA InP MSCs, F360-InP:Cl MSCs, and F399-InP:Cl MSCs (Table 1). In:P composition ratios were determined as 1:0.54, 1:0.32, and 1:0.48, respectively. The ratio of 1:0.54 for 386-InP MSCs accords well with the chemical formula of $In_{37}P_{20}$[13]. In:Cl ratios were 1:0.38 for F360-InP:Cl MSCs, and 1:0.45 for F399-InP:Cl MSCs; this increase in the Cl content in F399-InP:Cl MSCs over F360-InP:Cl MSCs also matches well with the relative binding energies in XPS. Counting the charges of In as +3, of P as −3, and of Cl as −1, the ratios of positive to negative charge are 1:0.54, 1:0.45, and 1:0.63 for 386-MA InP MSCs, F360-InP:Cl MSCs, and F399-InP:Cl MSCs, respectively. F360-InP:Cl MSCs have the most cationic composition. The charge inequality must have been neutralized by MA on surface. As a result, F360-InP:Cl MSCs should have a surface that is relatively more covered by MA than F399-InP:Cl MSCs. MA binding on surface should be rather labile; this explains well the instability of F360-InP:Cl MSCs and its thermal fragmentations that yielded the conversion to F399-InP:Cl MSCs. In contrast, F399-InP:Cl MSCs showed relatively well-balanced cation to anion charge ratio, which accounts for its notable chemical stability. F399-InP:Cl MSCs can be stored at RT for months without any noticeable change.

XRD patterns were obtained for 386-MA InP MSCs and InP:Cl MSCs (Fig. 5d). The polytwistane XRD pattern of 386-InP MSCs coincides with that of previously reported InP MSCs[25]. F360-InP:Cl MSCs showed an XRD pattern that was similar to that of 386-MA InP MSCs but quite broadened. However, F399-InP:Cl MSCs showed an XRD pattern that differs from those of 386-MA InP MSCs and F360-InP:Cl MSCs. F399-InP:Cl MSCs showed the

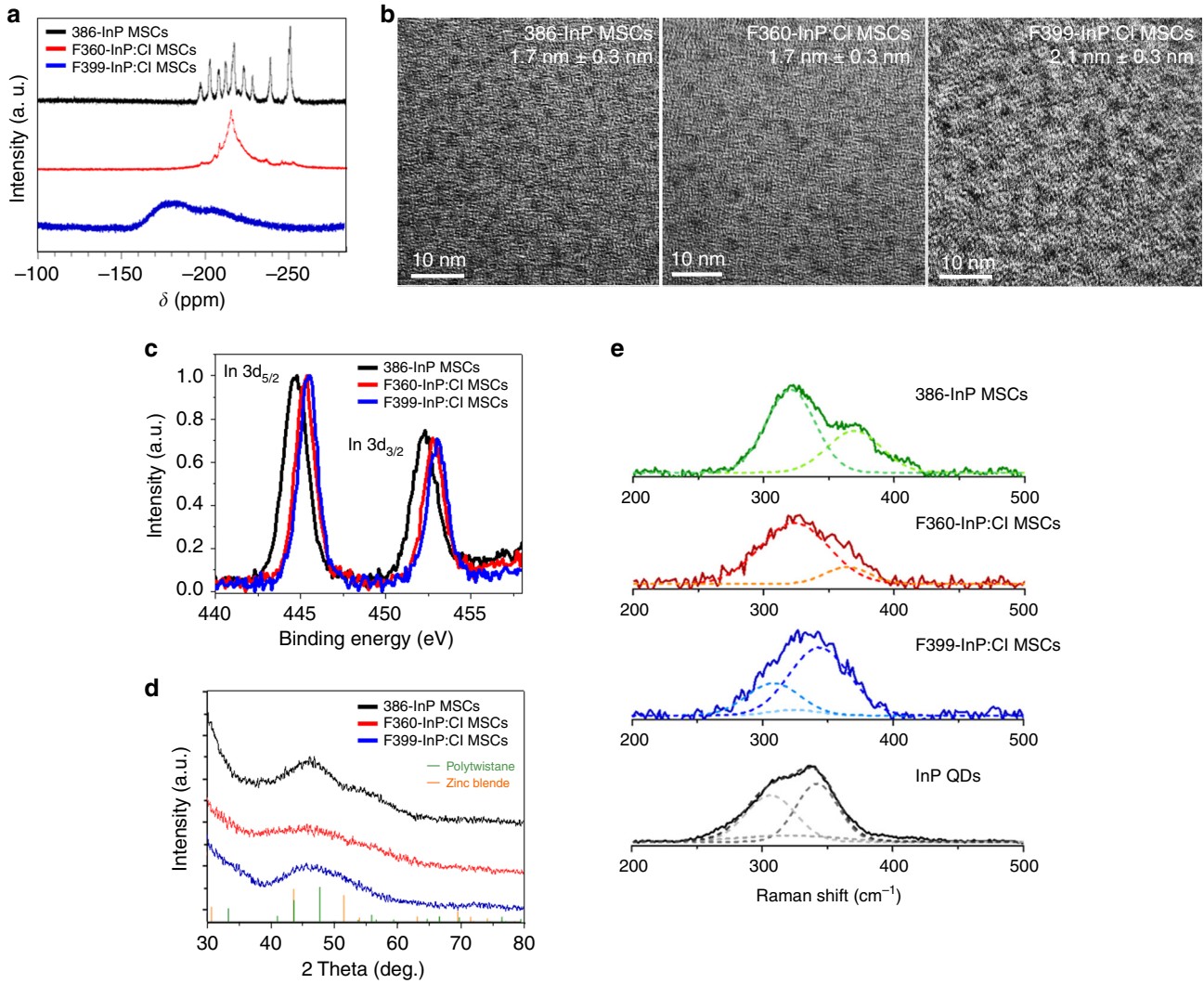

**Fig. 5 Structural characterizations of InP:Cl MSCs. a** $^{31}$P{$^{1}$H} NMR, **b** TEM images, **c** In 3d XPS, and **d** XRD patterns of 386-MA InP MSCs, F360-InP:Cl MSCs, and F399-InP:Cl MSCs. JCPDS diffraction file No. 32-0452 was used for zinc-blende InP. The XRD pattern of polytwistane phase was obtained from a previous publication[25]. **e** Raman spectra of 386-InP MSCs, F360-InP:Cl MSCs, F399-InP:Cl MSCs, and InP QDs with zinc-blend structure. All Raman spectra was fitted with two and three overlapping Gaussian peaks.

most prominent peak at 45.7°, which is shifted to a lower angle from those of 386-MA InP MSCs and F360-InP:Cl MSCs observed at 46.1°. F399-InP:Cl MSCs also showed significantly reduced scattering signals at ~55° when compared with the other two MSCs. These patterns of F399-InP:Cl MSCs are more consistent with that from zinc-blende structure.

Raman spectra were taken for 386-InP MSCs, F360-InP:Cl MSCs, F399-InP:Cl MSCs, and InP QDs, where, being similar to their XRD patterns, F399-InP:Cl MSCs showed distinguished spectrum from other MSCs (Fig. 5e). The Raman spectrum of 386-InP MSCs, identical to what has been previously reported by others[25], was fitted with two overlapping peaks at 320 cm$^{-1}$ and 368 cm$^{-1}$. These two peaks are significantly differentiated from typical Raman-active longitudinal optical (LO) and transverse optical (TO) peaks of InP QDs arising from a phonon confinement effect[25]. Considering the small size and highly assymetric polytwistane structure of 386-InP MSCs in contrast to a zinc-blende structure of InP QDs, the two Raman peaks are assigned as surface and interior vibrational modes arising from its molecular vibrational character[29]. F360-InP:Cl MSCs showed the similar Raman spectrum to 386-InP MSCs, which was also fitted with surface and interior vibrational peaks at 324 cm$^{-1}$ and 354 cm$^{-1}$,

respectively. Here, compared with 386-InP MSCs, the attenuated interior peak of F360-InP:Cl MSCs well supports the partial etching in the conversion for 386-InP MSCs to F360-InP:Cl MSCs. Moreover, the shifts and broadenings resulted from the minor structural change from 386-InP MSCs to F360-InP:Cl MSCs by the Cl incorporation.

On the other hand, the Raman spectrum of F399-InP:Cl MSCs was fitted with the three peaks at 310, 343, and 324 cm$^{-1}$ that are well matched with TO, LO, and surface optical (SO) Raman peaks of zinc-blende InP QDs, respectively (Fig. 5e)[30,31]. This significantly distinguished Raman spectral features of F399-InP:Cl MSCs from those of 386-InP MSCs and F360-InP:Cl MSCs manifest that the conversion from F360-InP:Cl MSCs to F399-InP:Cl MSCs accompanies structural change from polytwistane to zinc-blende structure. TEM measurements also showed that F399-InP:Cl MSCs are larger than F360-InP:Cl MSCs. It is suggested that relatively small MSCs such as 386-InP MSCs and F360-InP:Cl MSCs may prefer polytwistane structure to gain favorable bond energies between In and P atoms. The larger F399-InP:Cl MSCs should evolve to more zinc-blende structure, which is close to bulk thermodynamic equilibrium that possesses higher symmetry. In other words, an evolution

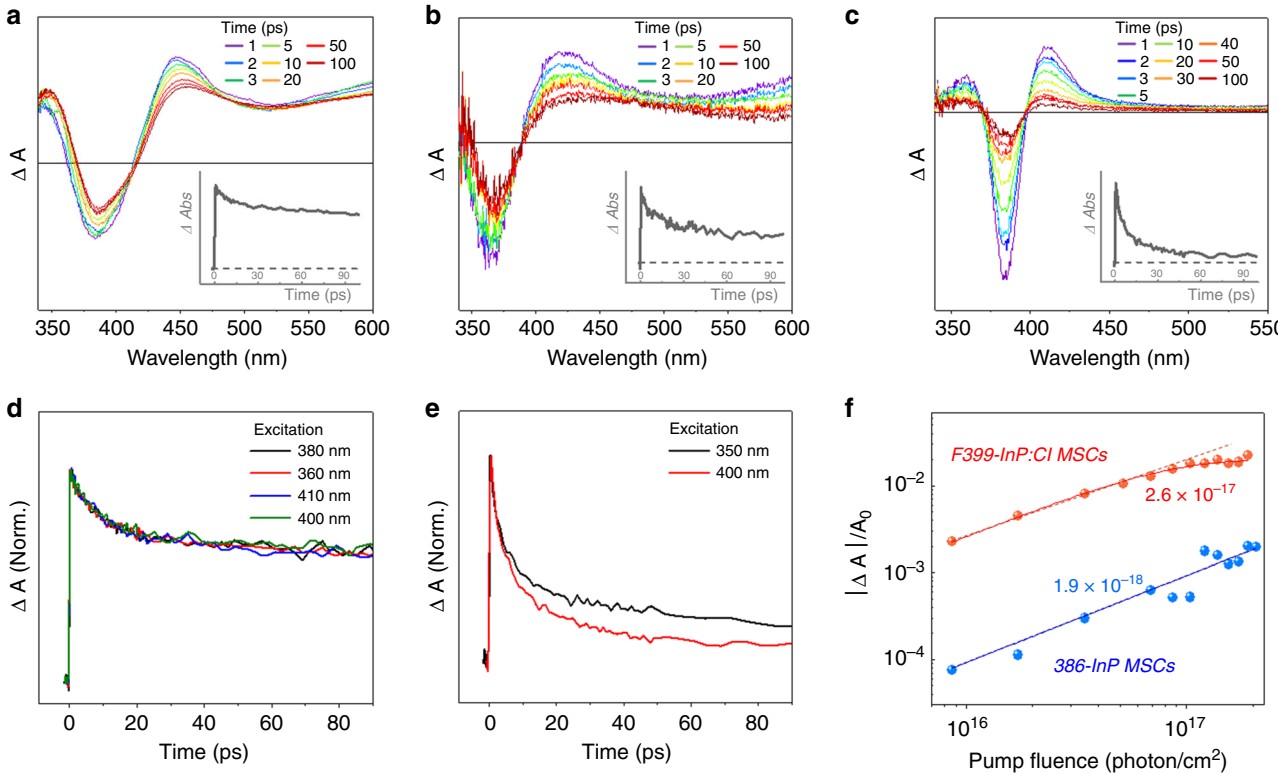

**Fig. 6 Optical characterizations of InP:Cl MSCs.** Transient absorption (TA) spectra of **a** 386-InP MSCs (inset: TA decay profile at 445 nm), **b** F360-InP:Cl MSCs (inset: TA decay profile at 410 nm), and **c** F399-InP:Cl MSCs in toluene (inset: TA decay profile at 410 nm). TA decay profiles of **d** 386-InP MSCs at 450 nm and **e** F399-InP:Cl MSCs at 410 nm with various excitation-wavelength. **f** Plot for normalized LEET bleach signals of 386-InP MSCs and F399-InP:Cl MSCs with the estimated absorption cross-section value $\sigma_{abs}$ (cm$^{-1}$).

from small-molecule like to more QD-like characters is observed for the evolution series from 386-InP MSCs to F360-InP:Cl MSCs and F399-InP:Cl MSCs.

**Optical characterizations of 386-InP MSCs, F360-InP:Cl MSCs, and F399-InP:Cl MSCs.** Steady-state absorption and emission spectra of 386-InP MSCs and InP:Cl MSCs were obtained (Supplementary Fig. 13). Similar to XRD and Raman results, 386-InP MSCs/F360-InP:Cl MSCs and F399-InP:Cl MSCs showed distinguished spectral features. The broad LEET and emission were observed in the former two MSCs, whereas F399-InP:Cl MSCs displayed the relatively sharp spectral features where the sharp and intense band at 420 nm and broad tailing over 440 nm can be assigned as QD-like band-edge and trap-associated emission, respectively.

The transient absorption (TA) spectra of F399-InP:Cl MSCs were also significantly distinguished from those of 386-InP MSCs and F360-InP:Cl MSCs (Fig. 6a–c). In F399-InP:Cl MSCs, the strong band-edge bleaching with narrow photoinduced absorption signal at lower-energy region was displayed, which is typical TA spectral features observed in well-defined QDs and describes a relaxation dynamics of exciton in quantum-confined systems[32–34]. On the other hand, the TA spectra of 386-InP MSCs and F360-InP:Cl MSCs showed different spectral features, the broad and long-lived photoinduced absorption signals in the lower-energy region of band-edge bleaching ~386 and 360 nm, respectively. Here, the significantly broad and long-lived ones of 386-InP MSCs and F360-InP:Cl MSCs are attributable to transitions from the lowest excited electronic state to higher excited electronic states, which is typically observed in molecular systems[35].

The excitation-wavelength-dependent TA measurements also showed contrasting results (Fig. 6d, e). The TA decay profiles of

F399-InP:Cl MSCs showed distinct excitation-wavelength dependency, where the accelerated initial decay with increased long-lived signals by higher energy excitation manifests the Auger and hot exciton/carrier trapping process in quantum-confined systems[36–38]. Unfortunately, although the TA data of F360-InP:Cl MSCs were not measured owing to its low photo-stability, the TA decay profiles of 386-InP MSCs show no excitation-wavelength dependent features, indicating that there is no Auger and trap-associated exciton dynamics. Therefore, these differences in the TA decay profiles well support the different electronic structures between 386-InP MSCs/F360-InP:Cl MSCs and F399-InP:Cl MSCs.

In addition, we measured the absorption cross-section ($\sigma_{abs}$) at the LEET, representing the ability to absorb a photon, with the absorption saturation method based on the state-filling model[38,39]. The $\sigma_{abs}$ values of 386-InP MSCs and F399-InP:Cl MSCs were estimated as $1.9 \times 10^{-18}$ cm$^{-2}$ and $2.6 \times 0^{-17}$ cm$^{-2}$, respectively (Fig. 6f), where F399-InP:Cl MSCs showed 14 times larger value than 386-InP MSC counterpart. This larger absorption cross-section at the LEET of F399-InP:Cl MSCs cannot be simply explained by the larger size of F399-InP:Cl MSC over 386-InP MSC, which indicates that their different electronic structures lead to the distinct difference in the absorption cross-section value as well as their optical properties.

These sharp contrasting features in 386-InP MSCs, F360-InP:Cl MSCs, and F399-InP:Cl MSCs can be comprehended with their different structures. Our NMR, XRD, and Raman results suggest that 386-InP MSCs and F360-InP:Cl MSCs possess the polytwistane structure with pseudo-$C_2$ symmetry. According to the previously reported crystal structure of 386-InP MSCs[13], the length and angle of In-P bonds show large fluctuations and In atoms are irregularly configured with the pseudo-octahedral and

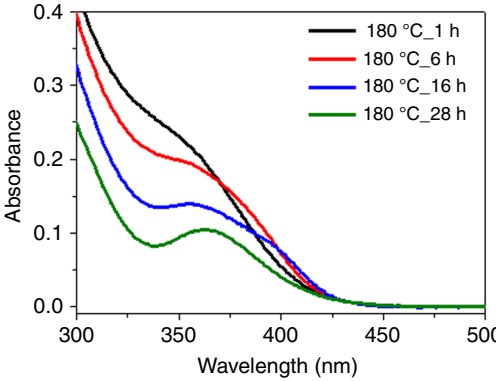

**Fig. 7 Synthesis of F360-InP:Zn MSCs.** UV − vis absorption spectra of aliquots taken during direct synthesis of F360-InP:Zn MSCs from In(Ac)$_3$, ODPA, (TMS)$_3$P, and Zn(SA)$_2$ precursors at 180 °C.

tetrahedral geometries in the polytwistane structure. Thus, in 386-InP MSCs and F360-InP:Cl MSCs, it is difficult to have quantum confinement along with their heterogeneous structures and they show the molecular-like character. On the other hand, for F399-InP:Cl MSCs, the conversion of F360-InP:Cl MSCs to F399-InP:Cl MSCs accompanies the structural evolvement to zinc-blende structure with an increase in size. Similar to InP QDs, the highly symmetric zinc-blende structure allows F399-InP:Cl MSCs to be quantum-confined systems, which results in the LO and TO Raman band with QD-like optical behaviors. Considering MSCs as intermediates between molecular monomers and QDs, it is quite expected to observe the evolution from unimolecular to colloidal-QD-like characters in MSCs.

**Syntheses of F360-InP:Zn MSCs, F408-InP:Zn MSCs, and F393-InP:Zn MSCs.** Three kinds of Zn-incorporated InP MSCs were synthesized: F360-InP:Zn MSCs, F408-InP:Zn MSCs, and F393-InP:Zn MSCs. In(Ac)$_3$ and ODPA were mixed by 1:2 ratio and heated to 300 °C in ODE to form In-ODPA complex. The solution was cooled to RT, then zinc stearate (Zn(SA)$_2$) and (TMS)$_3$P were added, and the solution was heated to 180 °C as stirring for 28 h to produce F360-InP:Zn MSCs. The In:Zn:ODPA:P ratio of 1:1:2:0.5 was used. After 1 h at 180 °C, a broad absorption peak around 360 nm emerged and became increasingly prominent over time (Fig. 7). After 28 h, the absorption spectrum remained unchanged and showed the LEET at 360 nm. F360-InP:Zn MSCs could be prepared using slightly different stoichiometries; however, typically resulted in lower reaction yields (Supplementary Fig. 15). In contrast, $T_R$ and the amount of ODPA critically affected the fate of product MSCs. When no ODPA was used, the growth was uncontrollable and yielded InP NPs (Supplementary Fig. 16). Reducing the amount of ODPA by quarter and elevating $T_R$ to 300 °C for 3.5 h yielded F408-InP:Zn MSCs, which had LEET at 408 nm (Supplementary Fig. 17). During the early stage of F408-InP:Zn MSC formation, F360-InP:Zn MSCs were transiently observed in the absorption spectra. Prompted by this observation, we sought conversion synthesis from F360-InP:Zn MSCs to F408-InP:Zn MSCs. When as-synthesized F360-InP:Zn MSCs were heated to 300 °C, F408-InP:Zn MSCs appeared and co-existed with F360-InP:Zn MSCs in 30 mins; complete mergence to F408-InP:Zn MSCs occurred in 2 h (Fig. 8a). The co-existing intermediates could be separated by ultra-centrifugation. F360-InP:Zn MSCs had the lower density (as a result of smaller inorganic core and larger number of surface-binding organic molecules) than F408-InP:Zn MSCs and could be isolated in the supernatant (Supplementary Fig. 18).

When the ODPA amount was increased (7.5% reduction from the representative F360-InP:Zn MSCs synthesis instead of the quarter reduction) at $T_R = 300$ °C for 12 h (Supplementary Fig. 19), a broad peak over 400 nm was observed during the initial stage. The broad peak was slightly blue-shifted and narrowed at 393 nm; this change indicates formation of F393-InP:Zn MSCs (Supplementary Fig. 19). The initial broad peak suggests co-existence of F408-InP:Zn MSCs and F393-InP:Zn MSCs. The conversion reaction from F408-InP:Zn MSCs to F393-InP:Zn MSCs was investigated (Fig. 8b). When as-synthesized F408-InP:Zn MSCs were kept stirring at 300 °C, the LEET blue-shifted from that of F408-InP:Zn MSCs to that of F393-InP:Zn MSCs. F393-InP:Zn MSCs showed a very narrow LEET at 393 nm (HWHM = 10 nm) with a second transition at 353 nm (Fig. 8b). PLE experiments confirmed the doublet absorption feature (Supplementary Fig. 20). The formation of F393-InP:Zn MSCs was independent of the pathways by which the starting F408-InP:Zn MSCs were prepared (i.e., direct synthesis or conversion synthesis) (Supplementary Fig. 21). The conversion from F360-InP:Zn MSCs to F408-InP:Zn MSCs appeared to be quantized, whereas the conversion from F408-InP:Zn MSCs to F393-InP:Zn MSCs showed a continuous blue-shift, which indicates the existence of numerous intermediate species between F408-InP:Zn MSCs and F393-InP:Zn MSCs. The amount of ODPA critically determined the conversion kinetics from F408-InP:Zn MSCs to F393-InP:Zn InP MSCs (Supplementary Fig. 22). Increase in the amount of ODPA facilitated faster conversion to F393-InP:Zn MSCs. Among InP:Zn MSCs, the final product F393-InP:Zn MSCs showed the narrowest LEET and the highest stability. F393-InP:Zn MSCs remained unchanged and did not grow into NPs at 300 °C even after days (Supplementary Fig. 23).

Unlike InP:Cl MSCs, InP:Zn MSCs could not be isolated when tried to prepare from 386-InP MSCs by conversions. InP:Zn MSCs grew more easily into NPs than InP:Cl MSCs did. The $T_R$ that can initiate structural change to 386-InP MSCs for conversion was typically too high to properly isolate InP:Zn MSCs before they grew rapidly to form NPs.

**Characterizations of F360-InP:Zn MSCs, F408-InP:Zn MSCs, and F393-InP:Zn MSCs.** Elemental analyses for InP:Zn MSCs were performed using inductively coupled plasma atomic emission spectrometry (Table 2). Equimolar amounts of In and Zn were used for all of the three kinds of InP:Zn MSCs. The elemental analyses revealed the Zn/In content ratios of 0.16 for F360-InP:Zn MSCs, 0.28 for F408-InP:Zn MSCs, and 0.23 for F393-InP:Zn MSCs. The quantized conversion from F360-InP:Zn MSCs to F408-InP:Zn MSCs may involve thermal fragmentations and rearrangements, where the fragmentations are associated with Zn-complex-mediated assemblies and the rearrangements accompanied by additional Zn incorporations increase the amount of Zn content. The conversion from F408-InP:Zn MSCs to F393-InP:Zn MSCs involved slight loss of Zn complexes (i.e., detachment of Zn(SA)$_2$ complexes). Owing to the phosphonate ligands, all samples had phosphorus-rich compositions.

The continuous transition from F408-InP:Zn MSCs to F393-InP:Zn MSCs suggests that the two MSCs have the same inorganic skeleton. To test this hypothesis, structural characterizations were performed using XRD and Raman spectroscopy. The XRD pattern of F360-InP:Zn MSCs resembles that of polytwistane 386-InP MSCs. In contrast, F408-InP:Zn MSCs, and F393-InP:Zn MSCs showed distinct peak at 44.0° and 54.8°, which cannot be related to either polytwistane or zinc-blende InP (Fig. 8c). F360-InP MSCs contain less Zn than F408-InP:Zn MSCs and F393-InP:Zn MSCs; this difference may have helped to preserve the polytwistane structure in F360-InP:Zn MSCs. In

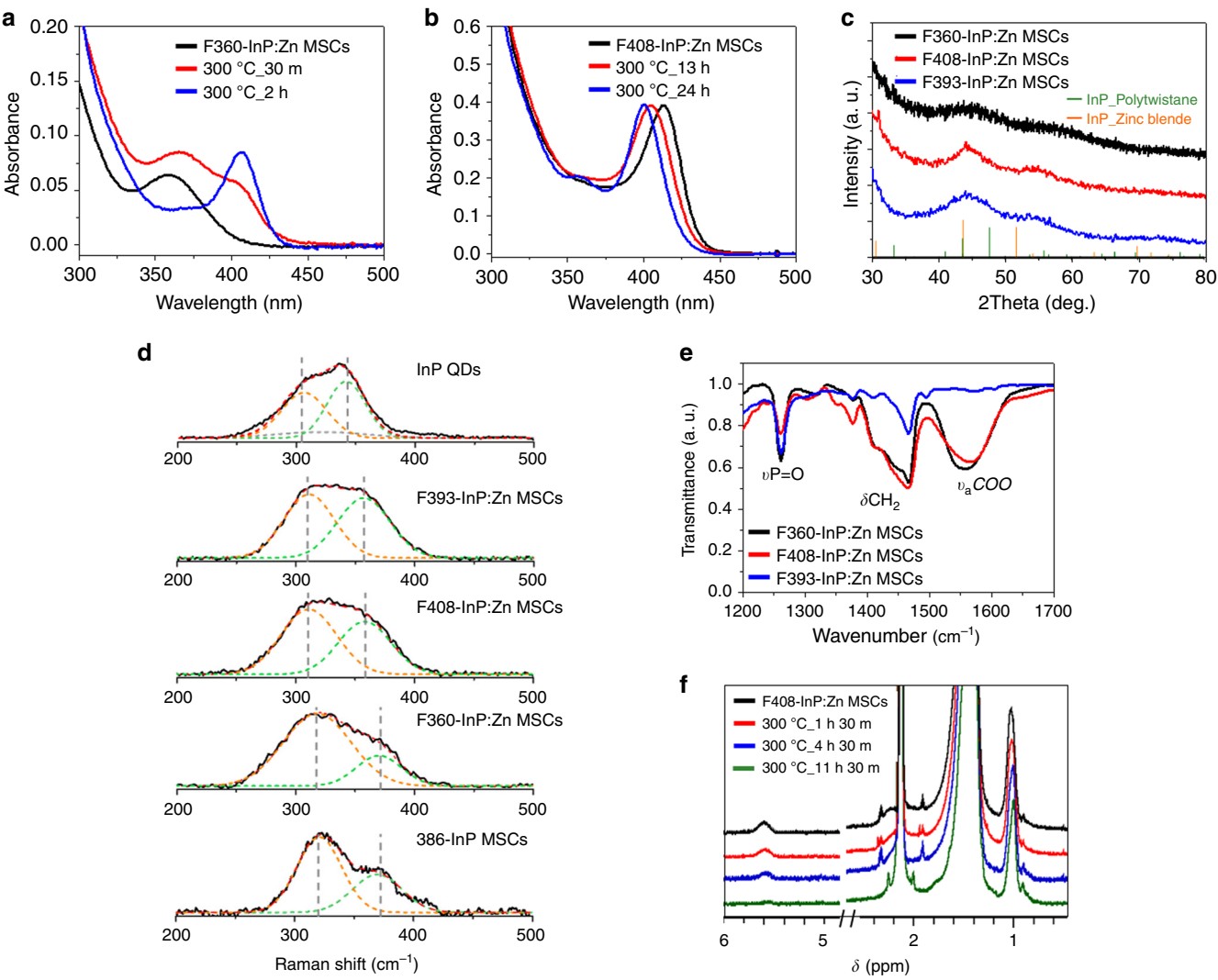

**Fig. 8 Conversion syntheses and characterizations of InP:Zn MSCs.** UV−vis absorption spectra of aliquots taken during **a** conversion synthesis of F408-InP:Zn MSCs from as-synthesized F360-InP:Zn MSCs and during **b** conversion synthesis of F393-InP:Zn MSCs from as-synthesized F408-InP:Zn MSCs. **c** XRD patterns of F360-InP:Zn MSCs, F408-InP:Zn MSCs, and F393-InP:Zn MSCs. JCPDS diffraction file No. 32-0452 was used for zinc-blende InP. The XRD pattern of polytwistane phase was obtained from a previous publication[25]. **d** Raman spectra of InP QDs with zinc-blend structure, F393-InP:Zn MSCs, F408-InP:Zn MSCs, F360-InP:Zn MSCs, and 386-InP MSCs. **e** FTIR spectra of F360-InP:Zn MSCs, F408-InP:Zn MSCs, and F393-InP:Zn MSCs. **f** [1]H NMR spectra of aliquots taken during conversion synthesis of F393-InP:Zn MSCs from F408-InP:Zn MSCs.

**Table 2 Chemical compositions (atomic %) of F360-InP:Zn MSCs, F408-InP:Zn MSCs, and F393-InP:Zn MSCs, obtained using ICP-AES.**

| MSC | In | P | Zn |
|---|---|---|---|
| F360-InP:Zn | 1.0 | 1.3 | 0.16 |
| F408-InP:Zn | 1.0 | 1.2 | 0.28 |
| F393-InP:Zn | 1.0 | 1.3 | 0.23 |

We report syntheses for two families of heterogeneous-atom-incorporated InP MSCs that have chlorine atoms or zinc atoms. Each series of MSCs could be prepared by sequential conversions. 386-InP MSCs could be converted to F360-InP:Cl MSCs, then to F399-InP:Cl MSCs. Similarly, F360-InP:Zn MSCs could be converted to F408-InP:Zn MSCs, then to F393-InP:Zn MSCs. As the conversion proceeded, the MSCs bore more QD-like character.

incorporation can induce lattice contraction and XRD peaks shift to larger angles[23].

F360-InP:Zn MSCs showed the Raman spectrum, which was fitted with two Gaussian peaks at 315 and 368 cm$^{-1}$ (Fig. 8d). These two Raman peaks are matched with those of 386-InP MSCs, which is consistent with the XRD analysis and well supports the polytwistane structure of F360-InP:Zn MSCs. The Raman bands of F408-InP:Zn MSCs and F393-InP:Zn MSCs were observed at the same positions, 310 and 355 cm$^{-1}$, which is in accordance with their XRD results and suggests that they have similar zinc-blende type structure. Here, compared to InP QDs, the shifted Raman bands of F408-InP:Zn MSCs and F393-InP:Zn MSCs to higher energy region well describe their contracted lattice structure by the Zn incorporation[40], which is well matched with their XRD analysis results. Moreover, the lower intensity of higher energy Raman band of F408-InP:Zn MSCs is attributed to the more Zn contents than F393-InP:Zn MSCs, which cause a peak broadening by inter-diffusion of Zn and enhanced lattice disorder[24].

contrast, F408-InP:Zn MSCs and F393-InP:Zn MSCs may have undergone radical structural change by incorporating a significant portion of Zn. The resultant structure may take In$_x$Zn$_y$P alloyed zinc-blende structure, where the substitutional Zn

Carboxylate and phosphonate ligands are available for F408-InP:Zn MSCs and F393-InP:Zn MSCs, so the surface ligands were characterized using IR and NMR spectroscopy. Fourier-transform infrared spectroscopy (FTIR) spectra were obtained for F360-InP:Zn MSCs, F408-InP:Zn MSCs, and F393-InP:Zn MSCs (Fig. 8e). F360-InP:Zn MSCs and F408-InP:Zn MSCs showed the strong characteristic peak at 1558 cm$^{-1}$, which corresponds to COO$^{-}$ asymmetric vibration of carboxylates. Surprisingly, the carboxylate peak was completely absent in the spectra of F393-InP:Zn MSCs. These results suggest that F393-InP:Zn MSCs are capped only by phosphonate ligands, whereas F360-InP:Zn MSCs and F408-InP:Zn MSCs are capped with both phosphonates and carboxylates. The continuous blue-shift in absorption from F408-InP:Zn MSCs to F393-InP:Zn MSCs should be surface change from mixed ligands to a solely phosphonate surface. To quantify the surface ligand populations in conversion from F408-InP:Zn MSCs to F393-InP:Zn MSCs, the analytical $^1$H NMR spectroscopic approach was used. As discussed earlier, alkene-containing HOA was used. F408-InP:Zn MSCs were prepared using OA instead of stearate (Supplementary Fig. 24). $^1$H NMR analysis was conducted after conversion to F393-InP:Zn MSCs at 300 °C. Aliquots were taken before and during the conversion until the complete conversion at 11.5 h. Each aliquot was rigorously purified by GPC, then used to obtain NMR spectra (Fig. 8f). Absorption spectra were also measured on samples that had been drawn during the conversion (Supplementary Fig. 24). Two characteristic peaks were monitored: δ at 5.59 ppm, which corresponds to the alkenyl protons in the bound OA, and δ at 1.02 ppm that represents terminal methyls in OA and phosphonate. As the F393-InP:Zn MSCs converted from F408-InP:Zn MSCs, the alkenyl proton peak decreased, and complete disappeared. These results prove the ligand-exchanging event during the conversion from F408-InP:Zn MSCs to F393-InP:Zn MSCs. The continuous blue-shift observed during the conversion is attributed to the surface ligand change. From the peak intensities of the two characteristic peaks, the number ratio of carboxylate to phosphonate bound onto F408-InP:Zn MSCs was deduced to be 1:5.9, which was eventually changed to solely phosphonate for F393-InP:Zn MSCs. In conversion from F408-InP:Zn MSCs to F393-InP:Zn MSCs, absorption and FTIR spectra were followed (Supplementary Fig. 25). Adding larger amount of ODPA significantly accelerate the conversion to F393-InP:Zn MSCs. IR spectra showed that the binding mode of phosphonate ligands changed more to divalent as the conversion proceeded. Narrowing of the P-O(H) vibration peaks between 1000 and 1200 cm$^{-1}$ was observed, which attributes to the change to deprotonated ODPA species from mixed ODPA species of (partially) protonated and deprotonated ODPAs. The deprotonated ODPA ligands should bind to the surface In atoms more strongly than protonated ODPAs, and this should make the final F393-InP:Zn MSCs to be very stable. Thermodynamically, the nature of the evolution can be considered as the enthalpy gain by exchanging the initial ligands to a stronger deprotonated ODPA ligand.

**Optical characterizations of InP:Zn MSCs and F397-ODPA InP MSCs**. Emission of F360-InP:Zn MSC was very broad and weak, whereas those of F408-InP:Zn MSC and F393-InP:Zn MSC were relatively sharp and strong with the quantum yields (QYs) of 2.5% and 2.0%, respectively (Supplementary Fig. 26a). PL decays were also investigated for InP:Zn MSCs (Supplementary Fig. 26b–d). For F360-InP:Zn MSC, similar PL decay profiles were observed at both emission peaks at 410 and 530 nm. On the other hand, F408-InP:Zn MSC and F393-InP:Zn MSC showed a significant different decay profiles between the PL maximum and

tailing regions. This contrasting PL features suggest that F360-InP:Zn MSC showed the emission feature more close to a molecular system, whereas the sharp and intense PL band and tail of F408-InP:Zn MSC and F393-InP:Zn MSC can be assigned as QD-like band-edge and trap-associated emissions, respectively. Here, F393-InP:Zn MSC showed the sharper band-edge-like emission with reduced tailing feature than F408-InP:Zn MSC. Particularly, at the position of band-edge-like emission, F408-InP:Zn MSC showed larger amplitude of longer decay component when compared with F393-InP:Zn MSC. In addition, F393-InP:Zn MSC has the lower PL QY than F408-InP:Zn MSC. These PL features indicate that the trap-associated emission is more enhanced in F408-InP:Zn MSC, which manifests smaller Zn content of F393-InP:Zn MSC than F408-InP:Zn MSC. This is also well in consistency with the fact the conversion from F408-InP:Zn MSCs to F393-InP:Zn MSCs involves loss of Zn complexes, and as the result F393-InP:Zn MSC has smaller Zn to In ratio than F408-InP:Zn MSC.

As F393-InP:Zn MSCs have only phosphonate for surface ligands, we prepared F397-ODPA InP MSC, which was previously reported by Gary et al. as InP MSCs with solely phosphonate surfaces[6], and investigated the effect of Zn incorporation on phosphonate-capped InP MSCs. Similar to the case of F393-InP:Zn MSC, XRD returned F397-ODPA InP MSC to be zinc-blende (Supplementary Fig. 27). Earlier optical studies compared the pair of 386-MA InP MSC and F399-InP:Cl MSC with pure InP MSC vs Cl-incorporated InP MSCs. Similarly, the pair of F397-ODPA InP MSC and F393-InP:Zn MSC was chosen here. Similarly, the pair of F397-ODPA InP MSC and F393-InP:Zn MSC was chosen here.

F397-ODPA InP MSC shows the very narrow LEET at 397 nm with additional electronic transition at 356 nm in the absorption spectrum (Fig. 9a). These spectral features of F397-ODPA InP MSCs display surprising resemblance with those of F393-InP:Zn MSCs, which showed the slightly broader and blue-shifted LEET at 393 nm with another electronic transition at 354 nm. Figure 9b exhibits the PL spectra for F397-ODPA InP MSCs and F393-InP:Zn MSCs, where the strong band-edge emission at 420 nm and trap-associated tailing over 445 nm were observed in both spectra. In addition, the TA spectra of F397-ODPA InP MSCs and F393-InP:Zn MSCs are strikingly similar (Fig. 9c, d), where the typical spectral features arising from the exciton relaxation dynamics in quantum-confined systems well appeared[32–34]. This high spectral similarity in the absorption, emission, and TA spectra manifests that both MSCs possess analogous zinc-blende structure, which leads to their QD-like character.

To compare structure-optical property relationship between 386-MA InP MSCs and F397-ODPA InP MSCs, 386-MA InP MSC showed the polytwistane structure and broad absorption feature at 386 nm, whereas F397-ODPA InP MSC was zinc-blende with narrow absorption feature at 397 nm. TA spectra of 386-MA InP MSC showed the broad and long-lived photoinduced absorption signals in the lower-energy region, which is attributable to transitions in molecular systems. To the contrary, F397-ODPA InP MSC showed the spectral features typically arising from the exciton relaxation dynamics in quantum-confined systems.

Compared with the weak emission tailing of F397-ODPA InP MSCs, F393-InP:Zn MSCs exhibited the intensified and broader PL emission over 445 nm. Moreover, the band-edge emission band of F393-InP:Zn MSCs is 1.3 times broader than that of F397-ODPA InP MSCs (Supplementary Figs. 28 and 29). These emissive features suggest that the incorporated Zn cations facilitate the carrier trapping process and intensifies the trap emission in F393-InP:Zn MSC. In the TA data, the broader photoinduced absorption signal in the lower-energy region of band-edge bleaching of F393-InP:Zn MSC with the longer-lived

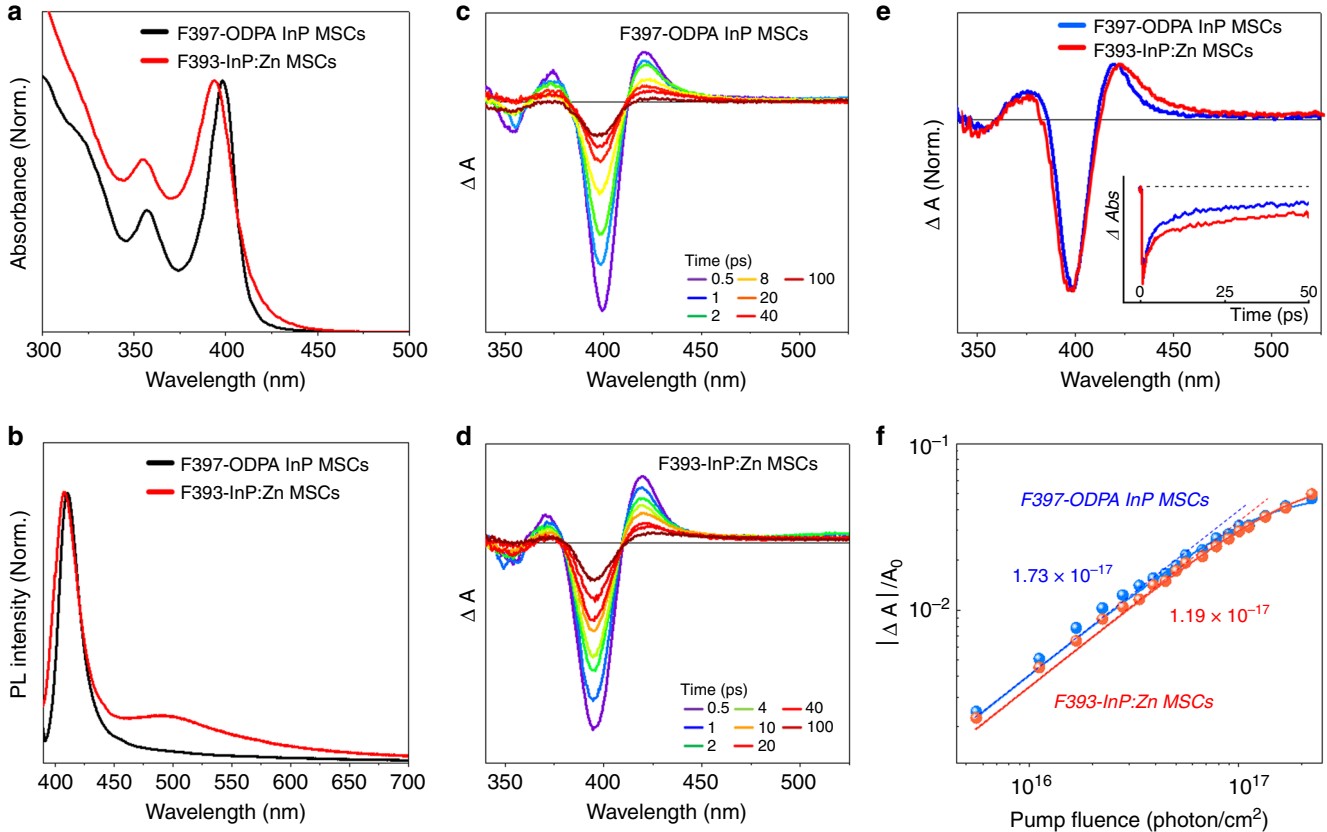

**Fig. 9 Optical characterizations of F393-InP:Zn MSCs and F397-InP MSCs.** Steady-state absorption and PL spectra of **a** F397-ODPA InP MSCs and **b** F393-InP:Zn MSCs. TA spectra of **c** F397-ODPA InP MSCs and **d** F393-InP:Zn MSCs. **e** Normalized TA spectra at 2 ps of F397-ODPA InP MSCs and F393-InP:Zn MSCs. (Inset: normalized TA decay profile of F397-ODPA InP MSCs at 397 and of and F393-InP:Zn MSCs at 393 nm.) **f** Plot for normalized LEET bleach signals for F397-ODPA InP MSCs and F393-InP:Zn MSCs with the estimated absorption cross-section values $\sigma_{abs}$ (cm$^{-1}$).

TA decay profile well depict the denser trap states by the Zn incorporation (Fig. 9e)[40]. Here, compared with F397-ODPA InP MSCs, the blue-shifted absorption spectrum of F393-InP:Zn MSCs may delineate the decreased effective size of "quantum-confined core", which suggests that the incorporated Zn cations may be mostly located at the "periphery" of the MSC. Such Zn incorporations can also reduce the symmetry of F393-InP:Zn MSCs, which yielded the smaller absorption cross-section value (Fig. 9f) and broad absorption and emission spectral features. We believe that F408-InP:Zn MSCs and F393-InP:Zn MSCs share a similar inorganic skeleton close to zinc-blende core but differ in Zn trap site density and that F393-InP:Zn MSCs and F397-ODPA InP MSCs may share a more similar core (both zinc-blende and electronically doublet absorption feature) but differ in Zn trap site existence. We believe the initial F360-InP:Zn MSCs to have the inorganic skeleton disparate from the rest of the InP:Zn MSCs series.

## Discussion

We report two families of Cl-incorporated InP MSCs and three families of Zn-incorporated InP MSCs. All of these MSCs could be directly synthesized from molecular precursors. 386-InP MSCs could be converted to F360-InP:Cl MSCs; this conversion required additional InCl$_3$, fatty acids, and mild heating. The conversion involved etching by in situ generated HCl, followed by rearrangements. F360-InP:Cl MSCs could be further converted to F399-InP:Cl MSCs at an elevated $T_R$. 386-InP MSCs and F360-InP:Cl MSCs had similar crystal structures, whereas F399-InP:Cl MSCs showed the transition from polytwistane to a structure similar to zinc-blende InP. F360-InP:Cl MSCs may share a similar inorganic

skeleton as 386-InP MSCs because F360-InP:Cl MSCs are from partially "etched" 386-InP MSCs. To the contrary, F399-InP:Cl MSCs have the volume almost two times larger than 386-InP MSCs and F360-InP:Cl MSCs and thus show the more-balanced In:P composition, which explains the crystal structure evolution to zinc-blende.

F360-InP:Zn MSCs could be converted to F408-InP:Zn MSCs, which could be further converted to F393-InP:Zn MSCs. F360-InP:Zn MSCs were thermally fragile, whereas F408-InP:Zn MSCs and F393-InP:Zn MSCs were quite stable. F393-InP:Zn MSCs showed an extreme stability against heat. The transition from F408-InP:Zn MSCs to F393-InP:Zn MSCs involved ligand change to divalent phosphonate. Emission of F360-InP:Zn MSC was very broad and weak, whereas those of F408-InP:Zn MSC and F393-InP:Zn MSC were relatively sharp and strong. F360-InP:Zn MSC had the PL decays that were independent of the emission wavelengths. To the contrary, F408-InP:Zn MSC and F393-InP:Zn MSC showed significant different decays between the PL maximum and tailing regions. F360-InP:Zn MSC showed the emission feature more close to a molecular system, whereas F408-InP:Zn MSC and F393-InP:Zn MSC exhibited QD-like band-edge and trap-associated emissions. F408-InP:Zn MSC showed the higher PL QY than F393-InP:Zn MSC and showed the larger amplitude at long decay component when compared with F393-InP:Zn MSC. These PL features indicate that the trap-associated emission is more enhanced in F408-InP:Zn MSC, which accords well with the fact that F408-InP:Zn MSCs contain larger Zn content than F393-InP:Zn MSC. Both F408-InP:Zn MSCs and F393-InP:Zn MSCs exhibited QD-like PL properties, however the characters were slightly different by the population of Zn trap sites.

For both InP:Cl and InP:Zn MSCs, evolutions from unimolecular-like to more QD-like characters were observed. However, structural identifications presented herein for each species were rather limited[16]. It is not simply because the evolutions involve many species but there are limited tools to directly elucidate the structures of MSCs. Characters of each species were extracted and compared by combining data from XRD, NMR, XPS, elemental analysis, and optical studies including TA and Raman. Nevertheless, it is far from pinpointing the structures of each species. Further structural identification is currently ongoing for the MSCs using synchrotron radiation, and hopefully will provide better understanding.

Our studies introduce multi-step sequential conversions among InP MSCs which evolved to be more QD-like as the conversions proceeded. Evolutions of MSCs are typically observed during early stages of nucleation and growth of QDs. It may not be a coincidence that MSCs become more and more QD-like as getting close to QDs. We are hopeful that further studies lead to mechanistic insight in QD formations and pave the way toward rational design of high-quality QD synthesis and development of new nanostructures that cannot be attained by conventional synthesis.

## Methods

**Chemicals**. Indium acetate (In(Ac)$_3$, 99.99%), indium chloride (InCl$_3$, 98%), myristic acid (HMA) (≥99%), oleic acid (OA) (≥99%), octadecylphosphonic acid (ODPA) (97%), chlorotrimethylsilane (TMS-Cl) (≥98%), zinc stearate (Zn(SA)$_2$) (technical grade), zinc oxide (99.99%), toluene (99.5%), hexanes (98.5%), 1-octadecene (ODE, 90%), squalane (≥95%), toluene-d8 (99 atom % D), deuterium oxide (99.9 atom % D), hydrochloric acid aqueous solution (35.0%), and phosphoric acid aqueous solution (85.0%) were purchased from Aldrich. Tris(trimethylsilyl)phosphine ((TMS)$_3$P) (95%) was purchased from SK chemicals. Bio-Beads S-X1 GPC medium was purchased from Bio-Rad Laboratories, Inc. All chemicals were used without further purification.

**Characterization**. TEM images and EDX were acquired using a JEOL JEM-2100 microscope operating at 200 kV. XRD experiments were performed using a Dmax2500/PC (Rigaku) diffractometer with Cu Kα radiation. Absorption spectra were measured using an Agilent 8453 UV–vis spectrophotometer. FTIR were performed using a Vertex 70 FTIR spectrometer with a resolution of 4 cm$^{-1}$ and averaging over 256 scans. XPS was performed using a VG Scientific ESCALAB 250 with monochromatic Al-Kα radiation of 1486.6 eV. $^1$H and $^{31}$P{$^1$H} NMR spectra were recorded on 300-MHz Bruker Avance spectrometers. For $^{31}$P{$^1$H} NMR experiments, a solution of phosphoric acid in deuterium oxide was used as an external reference and toluene-d8 was used as a solvent.

**Synthesis of 386-MA InP MSCs**. 386-MA InP MSCs were synthesized using a reported method[6]. First, 0.8 mmol of In(Ac)$_3$ and 2.9 mmol of HMA in a three-necked round-bottom flask containing 20 ml of ODE was degassed for 2 h at 110 °C, then 0.4 mmol of (TMS)$_3$P was added to the mixture and it was stirred for 2 h under N$_2$ atmosphere.

**Direct synthesis of F360-InP:Cl MSCs**. First, 0.5 mmol of InCl$_3$, 0.8 mmol of In (Ac)$_3$, and 2.4 mmol of HMA in a three-necked round-bottom flask containing 30 ml of ODE was degassed for 2 h at 110 °C, then cooled to 50 °C. A solution of 0.4 mmol of (TMS)$_3$P in 1 ml ODE was added to the mixture and it was stirred for 66 h under N$_2$ atmosphere.

**Conversion synthesis of F360-InP:Cl MSCs**. First, 0.5 mmol of InCl$_3$ in a three-necked round-bottom flask containing 10 ml of was degassed for 2 h at 110 °C then cooled to RT. Then 20 ml of as-synthesized 386-MA or OA InP MSCs solution was injected into the InCl$_3$ solution at RT and the mixture were heated to 80 °C. Aliquots of the solution were taken during the reaction until it was completely converted to F360-InP:Cl MSCs in ~18 h.

**Direct synthesis of F399-InP:Cl MSCs**. First, 0.65 mmol of InCl$_3$, 0.65 mmol of In(Ac)$_3$, and 1.95 mmol of HMA in a three-necked round-bottom flask containing 30 ml of ODE was degassed for 2 h at 110 °C, then a solution of 0.4 mmol of (TMS)$_3$P in 1 ml ODE was added to the mixture at 110 °C and it was stirred for 1 h under N$_2$ atmosphere.

**Conversion synthesis of F399-InP:Cl MSCs**. These MSCs were synthesized in the same way as F360-InP:Cl MSCs, except that as-synthesized 386-MA or OA InP

MSCs solution. The mixture solution of InCl$_3$ and as-synthesized 386-MA or OA InP MSCs were heated to 110 °C and stirred for 1 h.

**Direct synthesis of F360-InP:Zn MSCs**. First, 0.6 mmol of In(Ac)$_3$ and 1.2 mmol of ODPA were prepared in 12 ml ODE. The solution was degassed for 2 h at 110 °C and heated for 2 h at 300 °C, then the solution was degassed again for 2 h at 110 °C. In another pot, a solution of 0.6 mmol of Zn(SA)$_2$ in 10 ml ODE was degassed for 2 h at 110 °C, then transferred into the indium phosphonate solution. Then 0.3 mmol of (TMS)$_3$P in 2 ml ODE was injected into the mixture at RT and the solution was heated to 180 °C. Aliquots were taken during synthesis to confirm the end of the reaction (~28 h).

**Direct synthesis of F408-InP:Zn MSCs**. F408-InP:Zn MSCs were synthesized using molecular precursors in almost the same way as F360-InP:Zn MSCs. First, 0.6 mmol of In(Ac)$_3$ and 0.9 mmol of octadecylphosphonic acid were prepared in 12 ml ODE. The solution was degassed for 2 h at 110 °C and heated for 2 h at 300 °C, then degassed again for 2 h at 110 °C. In another pot, a solution of 0.6 mmol of Zn (SA)$_2$ in 10 ml ODE was degassed for 2 h at 110 °C then transferred into indium phosphonate solution. Then 0.3 mmol of (TMS)$_3$P in 2 ml ODE was injected into the mixture at RT and the solution was heated to 300 °C Aliquots were taken during synthesis to confirm the end of the reaction. After ~3 h, the absorption peak at 408 nm emerged.

**Conversion synthesis of F408-InP:Zn MSCs**. As-synthesized F360-InP:Zn MSCs solution was heated to 300 °C; F408-InP:Zn MSCs could be obtained after 2 h.

**Direct synthesis of F393-InP:Zn MSCs**. F393-InP:Zn MSCs were synthesized from molecular precursors in a manner similar to F360-InP:Zn MSCs. First, 0.6 mmol of In(Ac)$_3$ and 0.11 mmol of ODPA were prepared in 12 ml ODE. The solution was degassed for 2 h at 110 °C and heated for 2 h at 300 °C, then degassed again for 2 h at 110 °C. In another pot, a solution of 0.6 mmol of Zn(SA)$_2$ in 10 ml ODE was degassed for 2 h at 110 °C then transferred into the indium phosphonate solution. Then 0.3 mmol of (TMS)$_3$P in 2 ml ODE was injected into the mixture at RT and the solution was heated to 300 °C. Aliquots were taken during synthesis to confirm the end of the reaction. After ~10 h, the absorption peak at 393 nm emerged.

**Conversion synthesis of F393-InP:Zn MSCs**. As-synthesized F408-InP:Zn MSCs solution was heated to 300 °C; F393-InP:Zn MSCs were completely synthesized and after 24 h.

**Synthesis of F397-ODPA InP MSCs**. F397-ODPA InP MSCs were synthesized using a reported method[6]. About 0.32 mmol of In(Ac)$_3$ and 0.58 mmol of ODPA were added to a three-necked round-bottom flask containing 6 ml squalane. The mixture was stirred for 2 h at 120 °C under vacuum condition, then heated to 370 °C under N$_2$ atmosphere. A solution of 0.4 mmol of (TMS)$_3$P in 0.5 ml squalane was injected into the mixture at ~250 °C as the temperature of the solution was rising. When the temperature reached 370 °C, the solution should be cooled.

**Purification of InP:Cl MSCs and InP:Zn MSCs**. For characterization, all MSCs were purified using GPC. The preparative GPC column was packed inside a glove box. In total, 4 g of Bio-beads were swollen in anhydrous toluene overnight, then transferred into a glass column. The as-synthesized InP or Zn-alloyed InP MSCs in ODE were transferred into the glove box and a portion (<1 ml) of the InP MSCs sample was then injected into the column with ~50 ml of anhydrous toluene as the eluent, and all the purified samples were collected. The purification of the MSCs using GPC was repeated at least five times and confirmed using $^1$H NMR spectroscopy.

**Steady-state absorption and photoluminescence measurements**. Steady-state absorption spectra were recorded using a UV/Vis spectrometer (Cary5000, Varian). Steady-state photoluminescence (PL) of Nanocluster in degassed toluene was measured using a fluorescence spectrophotometer (F-7000, Hitachi).

**Raman measurement**. Raman spectra were acquired using a LabRam Aramis Raman microscope with Horiba Synapse Front-illuminated CCD detector (Horiba Jovin Yvon). The instrument was calibrated daily to the 520 cm$^{-1}$ Raman line of a silicon standard excited by the 325-nmline of a He-Cd laser, and the 532-nm of an ND:Yag lases. Spectra reported in the main text were acquired using the 325-nm He-Cd laser. Samples were prepared by drop-casting nanocluster colloids onto silicon substrates; measurements were made in backscattering geometry.

**TA measurement**. The femtosecond time-resolved TA (fs-TA) spectrometer consisted of Optical Parametric Amplifiers (Palitra, Quantronix) pumped by a Ti: sapphire regenerative amplifier system (Integra-C, Quantronix) operating at 1-kHz repetition rate and an optical detection system. The generated OPA pulses had a

pulse width of ~100 fs in the range 280–2700 nm, and were used as pump pulses. White light continuum (WLC) probe pulses were generated using a CaF$_2$ window (3 mm thick) by focusing of small portion of the fundamental 800 nm pulses, which was picked off by a quartz plate before entering the OPA. The time delay between pump and probe beams was carefully controlled by making the pump beam travel along a variable optical delay (ILS250, Newport). Intensities of the spectrally dispersed WLC probe pulses are monitored by a High Speed spectrometer (Ultrafast Systems). To obtain the time-resolved TA difference signal (ΔA) at a specific time, the pump pulses were chopped at 500 Hz and absorption spectra intensities were saved alternately with or without pump pulse. Typically, 4000 pulses excited the samples to obtain the fs-TA spectra at a particular delay time. To prevent polarization-dependent signals, the polarization angle between pump and probe beam was set at the magic angle (54.7°) using a Glan-laser polarizer with a half-wave retarder. Pump-probe experiments achieved cross-correlation full-width and half maximum < 200 fs. To minimize chirp, all reflection optics were used in the probe beam path. The measurements were conducted with 0.5-mm path length of flow quartz cell under a circulation condition with a degassed solvent. After the fs-TA experiments, we carefully checked absorption spectra of all compounds to determine whether degradation and photo-oxidation of samples had caused any artifacts.

**Reporting summary**. Further information on research design is available in the Nature Research Reporting Summary linked to this article.

## Data availability
The data that support the findings of this study are available from the corresponding author upon reasonable request. The structural figure (Fig. 4a) was drawn using reported crystallographic information file (https://doi.org/10.1021/jacs.5b13214) (ref. [13]).

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

## Acknowledgements
This research at POSTECH was supported by Creative Materials Discovery Program through the National Research Foundation of Korea (NRF) funded by Ministry of Science and ICT (NRF-2019M3D1A1078302) and by NRF grant funded by the Korea government (MSIT) NRF-2019R1A2C2084013. The work at Yonsei University was supported the National Research Foundation of Korea (NRF) grant funded by the Korea government (MEST) (NRF-2016R1E1A1A01943379). This work was also supported by SAIT, Samsung Electronics Co., Ltd.

## Author contributions
D.K. and S.K. directed the study. S.K. and Y.K. conceived the idea and designed experiments. Y.K., E.L., A.A, G.B., and J.K. performed the syntheses and characterizations; J.O., S.H.L, and D.K performed time-resolved spectroscopy and Raman characterizations. All authors contributed to data analysis and writing and editing of the manuscript.

## Competing interests
The authors declare no competing interests.
