## [Peer Review File · Nature Communications]

Reviewers' comments:

Reviewer #1 (Remarks to the Author):

This report from Kim, Kim, and coworkers details the synthesis and interconversion of a family of indium phosphide clusters related to a structurally characterized intermediate (386-InP MSC) previously identified by Cossairt and coworkers. This report details how surface chemistry, in particular the introduction of chloride, and the incorporation of zinc into the cluster lattice alters the optical and structural properties of the resulting clusters. All of the materials are carefully characterized by NMR, XRD, Raman, and composition analysis. The connection between all of the reactions and structures is relatively clear, despite the complicated nature of the overall system and the nuance of the individual reactions. Overall the work paints a picture of the versatility of InP cluster synthesis and interconversion and suggests that surface chemistry and doping are tunable parameters in altering the structure and properties of these clusters. I believe the work merits publication. Based on the description of papers suitable for Nature Communications, which specifies the journal as a venue for comprehensive, rigorous and elegant studies that are of specialist interest, I think this study fits the bill. A few minor comments are suggested upon revision prior to publication as noted below.

1. Page 3: The statement "MSCs that host heterogeneous atoms have not been studied in detail" is not clear to me. Specifically the meaning of the word heterogeneous seems ambiguous. Cluster doping - the incorporation of exogenous cations, for example, has significant precedent:

<https://pubs.acs.org/doi/abs/10.1021/acsnano.6b03348>,

<http://pubs.acs.org/doi/10.1021/acs.chemmater.7b03075>,

<https://pubs.acs.org/doi/10.1021/acs.chemmater.8b01482>, are just a few examples.

2. Page 8: The abbreviation "OAs" is used - I think the authors are reviewing to oleic acid here. Elsewhere they have used "OA" to refer to that species. This typo should be corrected. Also, it would help to distinguish between oleate (deprotonated) and oleic acid in the case of these abbreviations. HOA and OA for example.

3. EDX appears to be the primary composition method used in this manuscript. Was the EDX acquired on the TEM samples? It seems surprising to me that the clusters would survive under the harsh beam conditions needed to acquire EDX analysis. It is more typical to assess compositions by a method like ICP-MS (or OES/AES). Do those data compare favorably with the observed EDX numbers? Overall the reported values seem reasonable and consistent with what others have seen, but I am surprised the data are reliable for such fragile objects.

4. What was the rationale for adding ODPA in the synthesis of the zinc containing clusters? This seems like a non-intuitive choice and as a reader I would appreciate understanding what the authors were thinking there. What happens when no ODPA is used? Given the precedent for ODPA-capped InP clusters reported in reference 6, do the authors have controls that help us understand how much of the optical properties come from ligand effects versus the incorporation of zinc?

Reviewer #2 (Remarks to the Author):

In this manuscript, the authors reported the synthetic pathways of doped InP clusters. By introducing Cl atom or Zn atom either in situ or in post synthesis step, two families of InP clusters InP:Cl and InP:Zn with different size have been prepared. Furthermore, in an individual family, the authors observed the conversion of clusters between different sizes. Although, the authors did present a large amount of results in this manuscript, the novelty is still missing. This manuscript looks more like a following work of the one published on JACS in 2016 (JACS, 2016, 138, 1510). Besides, the characterization is not complete. For example, what is the actual crystal structures of the new InP clusters? How does the Cl or Zn bind with clusters, on the surface or in the cluster? Is the method present here a general method and does it work for other cluster system besides InP? Therefore, with these concerns in my mind, I don't recommend to publish this manuscript on Nature Communication.

Reviewer #3 (Remarks to the Author):

Manuscript NCOMMS-19-31129 entitled "Evolution Series from Uni-Molecule-like to Colloidal-Quantum-Dot like Characters in Chlorine or Zinc Incorporated Indium Phosphide Magic Size Clusters" by Dongho Kim, Sungjee Kim, et al, presents the syntheses of InP MSC-386, Cl-incorporated InP MSC-360, Cl-incorporated InP MSC-399, Zn-incorporated InP MSC-360, Zn-incorporated InP MSC-408, and Zn-incorporated InP MSC-393. For the Cl incorporated MSCs, they were synthesized directly or via "sequential conversion" reactions with InP MSC-386. For the Zn incorporated MSCs, they were synthesized directly or via "sequential conversion" reactions with Zn-incorporated InP MSC-360. The evolution from Zn-incorporated InP MSC-408 to Zn-incorporated InP MSC-393 was claimed. These experimental results seem to be interesting.

Also, the authors claimed, based on XRD and Raman, that the structures of Cl-incorporated InP MSC-399, Zn-incorporated InP MSC-408, and Zn-incorporated InP MSC-393 are close to the QD zinc-blende structure. The structures of Cl-incorporated InP MSC-360, Zn-incorporated InP MSC-360 are similar to that of InP MSC-386. The structural studies do not seem to be convincing.

The authors concluded that "Our findings highlight the mechanistic insight that the growth of QDs can be understood as multi-step sequential conversions through different MSCs or evolutions of MSCs." The key conclusion does not seem to be connected with the experimental results.

If the authors provide the evolution pathways in a convincing and logical way, from InP MSC-386 to Cl-incorporated InP MSC-360 and to Cl-incorporated InP MSC-399, and from Zn-incorporated InP MSC-360 to Zn-incorporated InP MSC-408 and to Zn-incorporated InP MSC-393, the study might be publishable.

1. What is the composition-structure-property relationship between Cl-incorporated InP MSC-360 and

Cl-incorporated InP MSC-399?

2. What is the composition-structure-property relationship for Zn-incorporated InP MSC-360, Zn-incorporated InP MSC-408, and Zn-incorporated InP MSC-393?
3. Is it possible to provide the very experimental evidence on the transformation of Cl-incorporated InP MSC-399 to QDs, and Zn-incorporated InP MSC-408 to Zn-incorporated InP MSC-393 to QDs?
4. On Page 17, the authors wrote for Figure S18 that the “a continuous blue-shift indicates the existence of numerous intermediate species between Zn-incorporated InP MSC-408 and Zn-incorporated InP MSC-393”. What is the nature of the evolution? The authors claimed that the two clusters have different compositions.
5. On Page 21, the authors should provide in depth explanations on how to control the Zn atom to be on the surface, and the definition of “periphery” (does not affect the absorption peak position) vs “quantum-confined core” (does affect the absorption peak position). What is the composition difference between ODPA-capped InP MSC-397 and Zn-incorporated InP MSC-393? For Zn-incorporated InP MSC-360, 408, and 393, do they have different cores?
6. What is the composition-structure-property relationship for MA-capped InP MSC-386 (exhibiting one absorption singlet) and ODPA-capped InP MSC-397 (displaying one absorption doublet)?
7. For the InP MSC-386 to Cl-incorporated InP MSC-360 evolution shown in Figure 1c at 80 °C, the authors attributed the absorption peaking at 416 nm as one intermediate species (416-IS). The half width at half-maximum (HWHM) of “416-IS” is 20 nm, the same as that of Cl-incorporated InP MSC-360. Can this intermediate be Cl-incorporated InP MSC-416?
8. For the evolution from InP MSC-386 to Cl-incorporated InP MSC-360 shown in Figure S3b at 50 °C and Figure S3c at 25 °C, the authors claimed (Lines 141 to 142 on Page 6) “intermediates which have not been properly detected”. Regarding the evolution from InP MSC-386 to Cl-incorporated InP MSC-360 at these different temperatures studied as shown in Figures 1c, S3b, and S3c, any explanation why the experimental observation on “intermediates” is influenced by the temperatures?
9. For the Page 9 equation, any numbers can be assigned to x ?
10. For the Page 9 equation, if $y = 0$, the reaction becomes an anionic exchange reaction between P and Cl. Any explanation?
11. All the figures should be improved, including the color usage, the legend font size, and dpi. For example, the Figure 1 legend font size is far too small.
12. The format of the refs should be edited according to the magazine required format.

Response to reviewer 1

Comment 1: Page 3: The statement "MSCs that host heterogeneous atoms have not been studied in detail" is not clear to me. Specifically the meaning of the word heterogeneous seems ambiguous. Cluster doping - the incorporation of exogenous cations, for example, has significant precedent: <https://pubs.acs.org/doi/abs/10.1021/acsnano.6b03348>, <http://pubs.acs.org/doi/10.1021/acs.chemmater.7b03075>, <https://pubs.acs.org/doi/10.1021/acs.chemmater.8b01482>, are just a few examples.

Our response: We thank the reviewer for the helpful comment. Per the comment, we have altered the text in the revised manuscript as shown below. We have also updated the references accordingly.

(Revised manuscript page 3, omission of what precedes) InP MSCs of different sizes or structures, and InP MSCs of different surface ligands have been reported.^{6,14,17} However, InP MSCs that host exogenous atoms have not been studied in detail. Cation exchange of InP MSCs with Cd ions has been reported, however the work mainly focused on the structural transformation to Cd₃P₂ clusters.¹⁸ Here, we report two InP MSC families: one that incorporates chlorine (Cl) atoms (InP:Cl MSCs), and one that incorporates zinc (Zn) atoms (InP:Zn MSCs). (Omission of what follows)

⁶ Gary, D. C., Terban, M. W., Billinge, S. J. L. & Cossairt, B. M. Two-Step Nucleation and Growth of InP Quantum Dots via Magic-Sized Cluster Intermediates. *Chemistry of Materials* 27, 1432-1441, doi:10.1021/acs.chemmater.5b00286 (2015).

¹⁴ Gary, D. C. et al. Single-Crystal and Electronic Structure of a 1.3 nm Indium Phosphide Nanocluster. *Journal of the American Chemical Society* 138, 1510-1513, doi:10.1021/jacs.5b13214 (2016).

¹⁷ Ning, J. & Banin, U. Magic size InP and InAs clusters: synthesis, characterization and shell growth. *Chemical communications* 53, 2626-2629, doi:10.1039/c6cc09778b (2017).

¹⁸ Stein, J. L. et al. Cation Exchange Induced Transformation of InP Magic-Sized Clusters. *Chemistry of Materials* 29, 7984-7992, doi:10.1021/acs.chemmater.7b03075 (2017).

Comment 2: Page 8: The abbreviation "OAs" is used - I think the authors are reviewing to oleic acid here. Elsewhere they have used "OA" to refer to that species. This typo should be corrected. Also, it would help to distinguish between oleate (deprotonated) and oleic acid in the case of these abbreviations. HOA and OA for example.

Our response: We thank the reviewer for the helpful suggestion. Per the suggestion, we used following acronyms in the revised manuscript to avoid confusions. We have also corrected the typo. HOA: oleic acid, OA: oleate, HMA: myristic acid, and MA: myristate

Comment 3: EDX appears to be the primary composition method used in this manuscript. Was the EDX acquired on the TEM samples? It seems surprising to me that the clusters would survive under the harsh beam conditions needed to acquire EDX analysis. It is more typical to assess compositions by a method like ICP-MS (or OES/AES). Do those data compare favorably with the observed EDX numbers? Overall the reported values seem reasonable and consistent with what others have seen, but I am surprised the data are reliable for such fragile objects.

Our response: We thank the reviewer for the insightful comment. Elemental analysis by ICP-MS or OES/AES methods typically requires acid digestions, and meanwhile phosphorous-containing compounds (especially InP) often evolve volatile phosphorous compounds (eg. PH_3). Technically, proper pre-oxidation and subsequent digestion should yield accurate phosphorous contents of InP MSCs by ICP-AES or by ICP-OES. However, in our cases, the values obtained multiple times by ICP-AES fluctuated significantly. On the other hand, our MSCs were quite stable under e-beam (200 kV) at least for a few minutes, and the EDX data were reproducible.

Comment 4: What was the rationale for adding ODPA in the synthesis of the zinc containing clusters? This seems like a non-intuitive choice and as a reader I would appreciate understanding what the authors were thinking there. What happens when no ODPA is used? Given the precedent for ODPA-capped InP clusters reported in reference 6, do the authors have controls that help us understand how much of the optical properties come from ligand effects versus the incorporation of zinc?

Our response: We thank the reviewer for the insightful comment. When no ODPA was used, zinc-containing clusters grew very rapidly to NPs before any clusters could be properly isolated. Figure R1 shows the control reaction which omitted ODPA in direct F393-InP:Zn MSC synthesis. As heating up the reaction mixture, broad absorption was observed at 200°C which quickly turned into NPs (Figure R1a). The final products were InP NPs of a few nm in size (Figure R1b).

Indeed, adding ODPA in the synthesis of zinc-containing clusters is non-intuitive. ODPA seems to be a strongly binding ligand to InP MSCs, and can stabilize certain InP MSCs as inhibiting the further growth to NPs. We used ODPA for InP:Zn MSCs simply because we were not able to synthesize Zn-incorporated InP MSCs without using ODPA.

Regarding the issue of how much of the optical properties come from ligand effects versus the incorporation of zinc, we believe that ODPA is involved in formation of the inorganic skeleton of MSCs which strongly governs the optical properties. In other words, we speculate that ODPA (as bound on MSC surfaces) made both F397-InP MSC (in reference 6) and our F393-InP:Zn MSC to share an inorganic skeleton which in turn shared similar optical properties such as the doublet features.

386-InP MSC can be converted into F397-InP MSC upon heating with ODPA addition. Figure R2 shows the absorption change during the conversion from 386-InP MSC to F397-InP MSC. The conversion is made via intermediates which shows the absorption peak at around 360 nm. It is noted that F397-InP MSC was formed after apparent dissolution of the intermediates at 360 nm and that the doublet optical feature appeared only after the dissolution of the 360 nm intermediates (Figure R2). This suggests that the inorganic skeleton of F397-InP MSC can be very different from that of 386-InP MSC. XRD pattern of F397-InP MSC is also very different from that of 386-InP MSC (Figure R3).

As for another control, we have performed experiments of adding small amounts of ODPA to 386-InP MSC. The reaction was monitored by absorption and ^{31}P NMR (Figure R4). Up to 10 equivalent ODPA addition, the absorption peak did not noticeably change. However, it slightly broadened in the UV region presumably due to the ODPA binding and partial etching as monitored by NMR. These results show that binding of ODPA on InP MSC does not alter the optical properties as significantly as the inorganic skeleton structural change.

Based on the additional experimental data, the revised manuscript has been also properly expanded as shown below.

(Revised manuscript page 16, omission of what precedes) In contrast, T_R and the amount of ODPA critically affected the fate of product MSCs. When no ODPA was used, the growth was uncontrollable and yielded InP NPs (Figure S16). Reducing the amount of ODPA by quarter and elevating T_R to 300°C for 3.5 h yielded F408-InP:Zn MSCs, which had LEET at 408 nm (Figure S17). (Omission of what follows)

(Revised manuscript page 21, omission of what precedes) Since F393-InP:Zn MSCs have only phosphonate for surface ligands, we prepared F397-ODPA InP MSC, which was previously reported by Gary *et al* as InP MSCs with solely phosphonate surfaces,⁸ and investigated the effect of Zn incorporation on phosphonate-capped InP MSCs. Similar to the case of F393-InP:Zn MSC, XRD returned F397-ODPA InP MSC to be zinc-blende structure (Figure S25). Earlier optical studies compared the pair of 386-MA InP MSC and F399-InP:Cl MSC to pure InP MSC vs. Cl-incorporated InP MSCs. Similarly, the pair of F397-ODPA InP MSC and F393-InP:Zn MSC was chosen here. (Omission of what follows)

Figure R1. (a) UV-vis absorption spectra of aliquots taken over time upon heating up to 300°C and (b) TEM image of the final sample for the reaction of indium acetate, zinc stearate, and $(\text{TMS})_3\text{P}$ without ODPA.

Figure R2. UV-vis absorption spectra of aliquots taken over time for the reaction of 386-MA InP MSCs and 50 equivalents of indium acetate, 150 equivalents of ODPA in ODE at 220°C.

Figure R3. XRD patterns of 386-InP MSCs and F397-InP MSCs.

Figure R4. Evolution of (a) UV-vis absorption and (b) ^{31}P NMR spectra of 386-OA InP MSCs with addition of ODPA of 1, 2, 3, 5, 7, and 10 equivalents, relative to 386-OA InP MSC in toluene- d_8 . (c) Blue box in (b) corresponding to chemical shift range between 0 ppm and 40 ppm and (d) red box in (b) corresponding to chemical shift range between -170 ppm and -270 ppm.

Response to reviewer 2

Comment: In this manuscript, the authors reported the synthetic pathways of doped InP clusters. By introducing Cl atom or Zn atom either in situ or in post synthesis step, two families of InP clusters InP:Cl and InP:Zn with different size have been prepared. Furthermore, in an individual family, the authors observed the conversion of clusters between different sizes. Although, the authors did present a large amount of results in this manuscript, the novelty is still missing. This manuscript looks more like a following work of the one published on JACS in 2016 (JACS, 2016, 138, 1510). Besides, the characterization is not complete. For example, what is the actual crystal structures of the new InP clusters? How does the Cl or Zn bind with clusters, on the surface or in the cluster? Is the method present here a general method and does it works for other cluster system besides InP? Therefore, with these concerns in my mind, I don't recommend to publish this manuscript on Nature Communication.

Our response: Admittedly, we do not have actual crystal structures of the new InP magic size clusters (MSCs). However, we do not believe that our works are like a following work of the one published on JACS in 2016 (JACS, 2016, 138, 1510). The 2016 JACS paper reported a crystal structure of 386-InP MSC using single crystal crystallography. 386-InP MSC is a single species. To the contrary, all our samples are a family of plural isomers. We have clearly stated it in the beginning of our manuscript, and it is also clearly shown in the nomenclatures that all the samples start with 'F' as representing 'family'. All our samples have the mass weight of tens of kDa. With the co-existence of isomers, it is practically impossible to grow single crystals suitable for crystallography. We believe that characterizations of each and every MSCs (and of each isomers to be precise) using single crystal crystallography is beyond the scope of our manuscript.

Our manuscript reports conversions among MSCs. We report two series of MSC series and the evolutions from uni-molecule-like to colloidal-QD-like characters observed as the conversions proceed. Early stage MSCs possess the low-symmetry polytwistane structures, and they show uni-molecular-like characters such as active inter-state transition in the excited state dynamics. Later stage MSCs exhibit more QD-like characters: crystal structure evolution to more zinc-blende toward the bulk thermodynamic equilibrium that possesses higher symmetry, LO and TO Raman bands, Auger and trap-associated exciton dynamics, and narrow photoinduced absorptions at the lower-energy region.

Regarding the issue "how the Cl or Zn binds with clusters, on the surface or in the cluster?", we believe that Cl and Zn reside in the surface and periphery of the MSCs. Upon incorporation of Cl ions, XPS spectra showed the In $3d_{5/2}$ peaks increased by 0.56 eV for F360-InP:Cl MSCs and by 0.77 eV for F399-InP:Cl MSCs when compared to that of 386-InP MSC. This large increase in the binding energies is indicative of the direct bindings of Cl to In atoms. Considering the valency of Cl ion, we believe it is reasonable to believe that Cl atoms exist on the surface of MSC binding directly to

adjacent In atoms. The case of InP:Zn MSC is less clear. F393-InP:Zn MSCs show the intensified and broader PL emission over 445 nm. In addition, the band-edge emission band of F393-InP:Zn MSCs is 1.3 times broader than that of F397-ODPA InP MSCs (Figure S26 and Figure S27). These emissive features suggest that the incorporated Zn cations facilitate the carrier trapping process and intensifies the trap emission in F393-InP:Zn MSC. In the TA data, the broader photoinduced absorption signal in the lower-energy region of band-edge bleaching of F393-InP:Zn MSC with the longer-lived TA decay profile well depict the denser trap states by the Zn incorporation (Figure 8e). Based on the optical properties, we speculate that the incorporated zinc cations may be mostly located at the periphery of the InP MSC.

Regarding the issue if our method present here is a general method and works for other cluster systems besides InP, unfortunately our method is not directly expandable to other cluster systems. However, other dopants could be incorporated into InP clusters using our method. For an example, InP:Sn MSC and InP:Mn MSC could be successfully obtained (See Figure R5 for absorption spectra of InP:Sn MSC and InP:Mn MSC). We have tried to extend our method to other cluster systems including GaP MSCs and InAs MSCs. In our attempts to synthesize GaP MSCs, the growth was uncontrollable and NPs were typically obtained (Figure R6). In our attempts to synthesize InAs MSCs using our method, uncontrolled growth or amorphous clusters of size 1-2 nm were obtained. Similar results were also reported by others (*Chem. Mater.* **2016**, *28*, 8119-8122). It might be possible to modify our method and extend it for other MSCs however we believe it to be beyond the scope of this manuscript.

Figure R5. UV-vis absorption spectra of (a) InP:Sn MSCs and (b) InP:Mn MSCs. InP:Sn MSCs were synthesized using In-ODPA complex, $(\text{TMS})_3\text{P}$, and tin stearate ($\text{Sn}(\text{St})_2$) by heating up to 300°C . InP:Mn MSCs were synthesized using In-ODPA complex, $(\text{TMS})_3\text{P}$, and manganese stearate ($\text{Mn}(\text{St})_2$) by heating up to 300°C .

Figure R6. (a) UV-vis absorption spectra of aliquots taken over time at 210°C and (b) TEM images of the final samples for the reaction of gallium acetylacetonate, myristic acid and (TMS)₃P.

Response to reviewer 3

General Comment: Manuscript NCOMMS-19-31129 entitled “Evolution Series from Uni-Molecule-like to Colloidal-Quantum-Dot like Characters in Chlorine or Zinc Incorporated Indium Phosphide Magic Size Clusters” by Dongho Kim, Sungjee Kim, et al, presents the syntheses of InP MSC-386, Cl-incorporated InP MSC-360, Cl-incorporated InP MSC-399, Zn-incorporated InP MSC-360, Zn-incorporated InP MSC-408, and Zn-incorporated InP MSC-393. For the Cl incorporated MSCs, they were synthesized directly or via “sequential conversion” reactions with InP MSC-386. For the Zn incorporated MSCs, they were synthesized directly or via “sequential conversion” reactions with Zn-incorporated InP MSC-360. The evolution from Zn-incorporated InP MSC-408 to Zn-incorporated InP MSC-393 was claimed. These experimental results seem to be interesting. Also, the authors claimed, based on XRD and Raman, that the structures of Cl-incorporated InP MSC-399, Zn-incorporated InP MSC-408, and Zn-incorporated InP MSC-393 are close the QD zinc-blende structure. The structures of Cl-incorporated InP MSC-360, Zn-incorporated InP MSC-360 are similar to that of InP MSC-386. The structural studies do not seem to be convincing. The authors concluded that “Our findings highlight the mechanistic insight that the growth of QDs can be understood as multi-step sequential conversions through different MSCs or evolutions of MSCs.” The key conclusion does not seem to be connected with the experimental results. If the authors provide the evolution pathways in a convincing and logic way, from InP MSC-386 to Cl-incorporated InP MSC-360 and to Cl-incorporated InP MSC-399, and from Zn-incorporated InP MSC-360 to Zn-incorporated InP MSC-408 and to Zn-incorporated InP MSC-393, the study might be publishable.

Our response: We thank the reviewer for the insightful comment. Due to the discrete nature of nanoclusters, size or composition of clusters cannot be continuously tuned and synthesized. As the result, it is practically impossible to continuously change the composition and elucidate the structure and properties to clearly map the composition-structure-property relationship as conventionally performed for bulk materials. In addition, the small size of nanoclusters makes the structural analysis extremely difficult because of the lack of many repeating units. By the size, our nanoclusters should have only a few repeating unit cells at best within any single cluster. Technically, single crystal (made of periodic superlattice of nanoclusters) crystallography is the ultimate way to elucidate the nanocluster structure. However, all our samples are a family of plural isomers as all our samples start with 'F' as representing 'family'. They have the mass weight of tens of kDa. With the co-existence of isomers, it is very difficult to grow single crystals of each MSCs (and of each isomers) suitable for crystallography. We had to resort to spectroscopic tools for the structural analysis of our nanoclusters, which only afforded limited structural information. As the reviewer pointed out, our structural analysis is rather limited and as the result the final conclusion sentence is admittedly an overstatement. We have altered the sentence to tone down the claim in the revised manuscript as shown below.

(Revised manuscript page 23, omission of what precedes) “Our findings highlight the mechanistic insight that MSCs which appear before the growth to QDs transform by multi-step sequential conversions of MSCs or evolutions of MSCs.” (Omission of what follows)

Comment 1: What is the composition-structure-property relationship between Cl-incorporated InP MSC-360 and Cl-incorporated InP MSC-399?

Our response: We thank the reviewer for the comment. The In:P:Cl compositions of F360-InP:Cl MSC and F399-InP:Cl MSC are 1:0.32:0.38 and 1:0.48:0.45, respectively. F399-InP:Cl MSC showed the increased P ratio to In than F360-InP:Cl MSC. F399-InP:Cl MSC is 2.1 nm in size and larger than 1.7 nm F360-InP:Cl MSC. Given the similar surface compositions for both MSCs (as both having the same surface ligands), the larger structure of F399-InP:Cl MSC must have afforded more balanced In:P ratio because of the increased number of inner P atoms. In:Cl ratios were 1:0.38 for F360-InP:Cl MSCs and 1:0.45 for F399-InP:Cl MSCs; this increase in the Cl content in F399-InP:Cl MSCs over F360-InP:Cl MSCs also matches well with the relative binding energies in XPS. Both XRD and Raman spectroscopy suggests that F360-InP:Cl MSC is structurally close to polytwistane, whereas F399-InP:Cl MSC is more matched with zinc-blende. F360-InP:Cl MSC possesses the polytwistane structure with pseudo- C_2 symmetry. The polytwistane structure has large fluctuations in the length and angle of In-P bonds, and In atoms in polytwistane structure are irregularly configured with the pseudo-octahedral and tetrahedral geometries. We conjecture that relatively small F360-InP:Cl MSC

takes polytwistane structure to gain favorable bond energies between In and P atoms. The larger F399-InP:Cl MSCs should evolve to more zinc-blende structure, which is close to bulk thermodynamic equilibrium that possesses higher symmetry. The composition-structure relationship strongly determines the optical properties. F360-InP:Cl MSCs is difficult to have quantum confinement along with their heterogeneous structures and they show the molecule-like character. F360-InP:Cl MSCs is non-emissive. TA spectra of F360-InP:Cl MSC shows the broad and long-lived photoinduced absorption signals in the lower-energy region of band-edge. The significantly broad and long-lived ones of F360-InP:Cl MSCs are attributable to transitions from the lowest excited electronic state to higher excited electronic states, which is typically observed in molecular systems. On the other hand, the highly symmetric zinc-blende structure allows F399-InP:Cl MSCs to be quantum-confined systems, which results in the LO and TO Raman band with QD-like optical behaviors. F399-InP:Cl MSCs displayed the relatively sharp emission spectral features where the sharp and intense band at 420 nm and broad tailing over 440 nm can be assigned as QD-like band-edge and trap-associated emission. F399-InP:Cl MSC also showed the strong band-edge bleaching with narrow photoinduced absorption signal at lower-energy region, which is typical TA spectral features observed in well-defined QDs and describes a relaxation dynamics of exciton in quantum-confined systems.

Comment 2: What is the composition-structure-property relationship for Zn-incorporated InP MSC-360, Zn-incorporated InP MSC-408, and Zn-incorporated InP MSC-393?

Our response: We thank the reviewer for the comment. The In:P:Zn compositions of F360-InP:Zn MSCs, F408-InP:Zn MSCs, and F393-InP:Zn MSCs are 1:1.3:0.16, 1:1.2:0.28, and 1:1.3:0.23, respectively. Due to the phosphonate ligands, all the MSCs have phosphorus-rich compositions. To differentiate P atoms in the inorganic InP structure and P atoms in the phosphonate ligands, we tried to strip the phosphonate ligands without altering the inorganic structure however it was never successful. According to the XRD and Raman data, F360-InP:Zn MSC has the structure close to polytwistane whereas F408-InP:Zn MSCs and F393-InP:Zn MSCs share an inorganic structure which can be assigned to a slightly distorted zinc-blende structure. Similar to the case of Cl incorporated InP MSCs, structural evolution from low symmetry polytwistane to high symmetry zinc-blende was observed.

Because of the limited composition information and also because of our intention to compare Zn incorporated InP clusters with ‘undoped’ InP clusters, we have compared optical properties of F393-InP:Zn MSC against F397-ODPA InP MSC. Both F393-InP:Zn MSC and F397-ODPA InP MSC have solely phosphonate for their ligands, which helped us to eliminate possible ligand effect. According to XRD data, both F393-InP:Zn MSC and F397-ODPA InP MSC also shared a similar inorganic skeleton. By comparing F393-InP:Zn MSC against F397-ODPA InP MSC, we reached a conclusion

that the incorporated Zn cations facilitate the carrier trapping process and intensifies the trap emission and that the zinc cations may be mostly located at the periphery of the MSCs.

Per the suggestion of investigation for composition-structure-property relationship for F360-InP:Zn MSC, F408-InP:Zn MSC, and F393-InP:Zn MSC, we have compared the optical properties. Absorption (the lowest energy electronic transition) half-bandwidths of F360-InP:Zn MSC, F408-InP:Zn MSC, and F393-InP:Zn MSC are respectively 26 nm, 13 nm, and 10 nm. Emission of F360-InP:Zn MSC is very broad and weak, whereas those of F408-InP:Zn MSC and F393-InP:Zn MSC are relatively sharp and strong with the quantum yields of 2.5% and 2.0%, respectively (Figure R7a). PL decays were also investigated for F360-InP:Zn MSC, F408-InP:Zn MSC, and F393-InP:Zn MSC. For F360-InP:Zn MSC, similar PL decay profiles were observed at both emission peaks at 410 and 530 nm (Figure R7b). On the other hand, F408-InP:Zn MSC and F393-InP:Zn MSC showed a significant different decay profiles between the PL maximum and tailing regions. This contrasting PL features suggest that F360-InP:Zn MSC showed the emission feature more close to a molecular system whereas the sharp and intense PL band and tail of F408-InP:Zn MSC and F393-InP:Zn MSC, showing a distinct different PL decay, can be assigned as QD-like band-edge and trap-associated emissions, respectively. Here, F393-InP:Zn MSC showed the sharper band-edge-like emission with reduced tailing feature than F408-InP:Zn MSC (Figure R7a). Particularly, at the position of band-edge-like emission, F408-InP:Zn MSC showed larger amplitude of longer decay component when compared to F393-InP:Zn MSC. In addition, F393-InP:Zn MSC has the lower PL quantum yield than F408-InP:Zn MSC. These PL features indicate that the trap-associated emission is more enhanced in F408-InP:Zn MSC, which manifests smaller Zn content of F393-InP:Zn MSC than F408-InP:Zn MSC. This is also well in consistency with the fact the conversion from F408-InP:Zn MSCs to F393-InP:Zn MSCs involves loss of Zn complexes (i.e., detachment of zinc stearate complexes), and as the result F393-InP:Zn MSC has smaller Zn to In ratio than F408-InP:Zn MSC. Therefore, the smaller Zn content of F393-InP:Zn MSC must have formed less trap sites than F408-InP:Zn MSC and showed the above-mentioned properties as nicely showing the composition-structure-property relationship.

Based on the additional experimental data, the revised manuscript has been also properly expanded as shown below.

(Revised manuscript page 20, omission of what precedes) From the peak intensities of the two characteristic peaks, the number ratio of carboxylate to phosphonate bound onto F408-InP:Zn MSCs was deduced to be 1:5.9, which was eventually changed to solely phosphonate for F393-InP:Zn MSCs.

Emission of F360-InP:Zn MSC was very broad and weak, whereas those of F408-InP:Zn MSC and F393-InP:Zn MSC were relatively sharp and strong with the quantum yields of 2.5% and 2.0%, respectively (Figure S25a). PL decays were also investigated for F360-InP:Zn MSC, F408-InP:Zn

MSC, and F393-InP:Zn MSC (Figure S25b, Figure S25c, and Figure S25d). For F360-InP:Zn MSC, the similar PL decay profiles were observed at both emission peaks at 410 and 530 nm. On the other hand, F408-InP:Zn MSC and F393-InP:Zn MSC showed a significant different decay profiles between the PL maximum and tailing regions. This contrasting PL features suggest that F360-InP:Zn MSC showed the emission feature more close to a molecular system whereas the sharp and intense PL band and tail of F408-InP:Zn MSC and F393-InP:Zn MSC, showing a distinct different PL decay, can be assigned as QD-like band-edge and trap-associated emissions, respectively. Here, F393-InP:Zn MSC showed the sharper band-edge-like emission with reduced tailing feature than F408-InP:Zn MSC. Particularly, at the position of band-edge-like emission, F408-InP:Zn MSC showed larger amplitude of longer decay component when compared to F393-InP:Zn MSC. In addition, F393-InP:Zn MSC has the lower PL quantum yield than F408-InP:Zn MSC. These PL features indicate that the trap-associated emission is more enhanced in F408-InP:Zn MSC, which manifests smaller Zn content of F393-InP:Zn MSC than F408-InP:Zn MSC. This is also well in consistency with the fact the conversion from F408-InP:Zn MSCs to F393-InP:Zn MSCs involves loss of Zn complexes, and as the result F393-InP:Zn MSC has smaller Zn to In ratio than F408-InP:Zn MSC. (Omission of what follows)

Figure R7. (a) PL spectra of F360-InP:Zn MSCs, F408-InP:Zn MSCs, and F393-InP:Zn MSCs. Time-resolved PL decay profiles of (b) F360-InP:Zn MSCs, (c) F408-InP:Zn MSCs, and (d) F393-InP:Zn MSCs.

Comment 3: Is it possible to provide the very experimental evidence on the transformation of Cl-incorporated InP MSC-399 to QDs, and Zn-incorporated InP MSC-408 to Zn-incorporated InP MSC-393 to QDs?

Our response: We thank the reviewer for the helpful suggestion. 386-InP MSC can be sequentially converted into F360-InP:Cl MSC, F399-InP:Cl MSC, and eventually into InP QDs, when the heat-up temperature is raised to 180°C or higher (instead of 150°C which is the typical reaction temperature for the conversion of 386-InP MSC to F399-InP:Cl MSC). For additional experiments, 56 equivalents of InCl₃ was added to as-synthesized 386-InP MSCs and the absorption change was monitored as the temperature elevated (Figure R8a). At the beginning, absorption of 386-InP MSC was observed. When the temperature reached 100°C, conversion to F360-InP:Cl MSC was only observed, which was followed by complete conversion to F399-InP:Cl MSC at 150°C. Upon further heating to 200°C, broad and featureless absorption was observed at 180°C and higher. The final product was collected and characterized as InP QDs. XRD returned the crystal structure as zinc-blende InP (Figure R8b) and their average size was measured as 7.50 nm by TEM (Figure R8c).

Contrary to the case of Cl-incorporated InP MSCs, F393-InP:Zn MSCs could not be converted into QDs by heating. F393-InP:Zn MSC has the notable thermal stability, and it does not lose any integrity upon continuous heating at 300°C for weeks. Further heating of F393-InP:Zn MSC to a higher temperature than 300°C eventually resulted in the complete dissolution of clusters. However, when F393-InP:Zn MSCs were heated with addition of extra cationic or anionic molecular precursors such as In(MA)₃ or (TMS)₃P, F393-InP:Zn MSCs could be further grown into InP QDs.

When In(MA)₃ were added into F393-InP:Zn MSCs at room temperature, the absorption peak red-shifted to 404 nm and the second transition disappeared (Figure R9a). Spectroscopic analysis suggests the bridging bondings between two oxygen atoms of an ODPA and surface In atoms on F393-InP:Zn MSCs was opened and inserted by an incoming In(MA)₃ molecule (Figure R10). FT-IR bridging mode of carboxylate indicates two oxygen atoms of an ODPA are coordinated to two different In atoms (Figure R11). A few In(MA)₃ were additionally attached to the surface of F393-InP:Zn MSCs, which dramatically reduced the thermal stability. When F393-InP:Zn MSCs were added by 70

equivalents of $\text{In}(\text{MA})_3$ and heated up, the red-shift of absorption peak and disappearance of the second transition were followed by further gradual red-shift. At 300°C , the absorption profile became broad and featureless showing the tail down to 660 nm. InP QDs of atypical shapes were obtained (Figure R9b). The destabilized cationic-precursor-adsorbed F393-InP:Zn MSCs were thought to have disassembled into monomers and fragments, which subsequently re-assembled and grew into the atypical InP QDs.

When the anionic precursor $(\text{TMS})_3\text{P}$ was added to F393-InP:Zn MSCs, the absorption did not change up to 60°C (Figure R9c). Further heating over 200°C yielded red-shift of the absorption, and the absorption became featureless at 270°C . The absorption was also significantly broadened with the tail reaching over 500 nm, which is indicative of InP QD growth. The final product was aggregated InP QDs (Figure R9d). The initial red-shift upon the addition of $(\text{TMS})_3\text{P}$ was due to the silylester reaction between $(\text{TMS})_3\text{P}$ and phosphonate and carboxylate ligands on F393-InP:Zn MSCs. The surface silylester reaction destabilized F393-InP:Zn MSCs. The destabilized F393-InP:Zn MSCs became an In precursor source as slowly releasing In molecular precursor as being decomposed. The extra $(\text{TMS})_3\text{P}$ added were reacted with the released In precursors and nucleated and grew into the final InP QDs.

Both cationic precursor $\text{In}(\text{MA})_3$ and anionic precursor $(\text{TMS})_3\text{P}$ induced fragmentations of F393-InP:Zn MSCs. However, the chemical yields of InP QD growth was notably different. InP QDs grown from the addition of anionic $(\text{TMS})_3\text{P}$ was significantly larger than the case of cationic addition of $\text{In}(\text{MA})_3$. The mechanisms of growth of InP QDs from InP MSCs or from combinations of MSCs and molecular precursors are very complicated, and they are still under investigations. We believe such studies including the above-mentioned results to be beyond the scope of this manuscript and will be reported at a forthcoming publication.

Based on the additional experimental data, the revised manuscript has been also properly expanded as shown below.

(Revised manuscript page 7, omission of what precedes) At $T_{\text{r}} > 180^\circ\text{C}$, quick conversion to F399-InP:Cl MSCs was followed by further growth forming NPs (Figure S6b and Figure S6c). Heat-up experiment at $T_{\text{r}} > 180^\circ\text{C}$ also yielded similar results with zinc-blende InP NPs of a few nm in size (Figure S7) (Omission of what follows)

Figure R8. (a) UV-vis absorption spectra of aliquots taken over time during transformation of QDs from F399-InP:Cl MSCs using as-synthesized 386-MA InP MSCs and 56 equivalents of InCl_3 , (b) XRD pattern and (c) TEM image of InP QDs transformed from F399-InP:Cl MSCs.

Figure R9. UV-vis absorption spectra of aliquots taken over time upon heating up to 300°C and TEM images of the final samples for the reaction of: (a,b) F399-InP:Zn MSCs and 70 equivalents of $\text{In}(\text{MA})_3$, (c,d) F399-InP:Zn MSCs and 70 equivalents of $(\text{TMS})_3\text{P}$.

Comment 4: On Page 17, the authors wrote for Figure S21 that the “a continuous blue-shift indicates the existence of numerous intermediate species between Zn-incorporated InP MSC-408 and Zn-incorporated InP MSC-393”. What is the nature of the evolution? The authors claimed that the two clusters have different compositions.

Our response: We thank the reviewer for the helpful comment. The nature of the evolution from F408-InP:Zn MSCs to F393-InP:Zn MSCs is surface ligand exchange. F408-InP:Zn MSCs have mixed surface ligands of stearate and octadecylphosphonate. The stearate is exchanged by additional octadecylphosphonate, and the final F393-InP:Zn MSCs have solely octadecylphosphonate on surface. Since octadecylphosphonate also has the phosphorous composition, the phosphorous composition from F408-InP:Zn MSCs to F393-InP:Zn MSCs should be dependent on the octadecylphosphonate surface density as well as the inorganic InP composition change. In addition, some stearates are being exchanged as coordinating a Zn ion (as zinc-stearate complexes) by octadecylphosphonate. Such exchange results in decrease in the composition of Zn in F393-InP:Zn MSCs. We also believe such ligand exchange accompanies surface structure reconstructing. As the result, we believe there are many intermediates in conversion from F408-InP:Zn MSCs to F393-InP:Zn MSCs.

Figure R10. Proposed binding mode of $In(MA)_3$ on the surface of F393-InP:Zn MSCs.

Figure R11. (a) UV-vis absorption and (b) FT-IR spectra of F393-InP:Zn MSCs and F393-InP:Zn MSCs reacted with 70 equivalent of $In(MA)_3$.

We have performed additional experiments of converting F408-InP:Zn MSCs to F393-InP:Zn MSCs by adding 0.925 or 0.875 equivalents of ODPA with respect to SA (ODPA: Octadecylphosphonate, SA: Stearate). Both absorption spectra and FT-IR spectra confirm that adding the larger amount of ODPA significantly accelerate the conversion to F393-InP:Zn MSCs (Figure R12). When 0.925 ODPA/SA ratio was used, carboxylate ligands were completely removed in 24 hrs (Figure R12a and Figure R12b). To the contrary, 0.875 ODPA/SA ratio needed 72 hrs for such conversion. During the exchange reaction, narrowing of the P-O(H) vibration peaks between 1000 and 1200 cm^{-1} was observed, which attributes to the change to deprotonated ODPA species from mixed ODPA species of (partially) protonated and deprotonated ODPA. All the deprotonated ODPA should bind to the surface In atoms more strongly than partially protonated ODPA, and this makes the final F393-InP:Zn MSCs to be very stable. ODPA is known to be a stronger binding ligand than SA or zinc stearate (*J. Phys. Chem. Lett.* **2011**, 2, 145–152). The nature of the evolution can be considered as the

Figure R12. (a) UV–vis absorption spectra and (b) FT-IR spectra of aliquots taken during direct synthesis of F393-InP:Zn MSCs from indium acetate, ODPA, $(\text{TMS})_3\text{P}$ and zinc stearate precursors at 300°C. The molar ratio of indium acetate to ODPA is 1:0.925. (c) UV–vis absorption spectra and (d) FT-IR spectra of aliquots taken during direct synthesis of F393-InP:Zn MSCs from indium acetate, ODPA, $(\text{TMS})_3\text{P}$ and zinc stearate precursors at 300 °C. The molar ratio of stearate to ODPA is 1:0.875.

enthalpy gain by exchanging the ligand to a stronger ODPA ligand.

Comment 5: On Page 21, the authors should provide in depth explanations on how to control the Zn atom to be on the surface, and the definition of “periphery” (does not affect the absorption peak position) vs “quantum-confined core” (does affect the absorption peak position). What is the composition difference between ODPA-capped InP MSC-397 and Zn-incorporated InP MSC-393? For Zn-incorporated InP MSC-360, 408, and 393, do they have different cores?

Our response: We thank the reviewer for the insightful comment. The definitions of “periphery” (does not affect the absorption peak position) and “quantum-confined core” (does affect the absorption peak position) are not physically characterized in our study. Rather, they are relative terms distinguishing periodic InP crystal core vs. peripheral inorganic regions incorporating Zn ions to explain the optical properties such as smaller absorption cross section value and broad absorption and emission spectral features of F393-InP:Zn MSCs in comparison to F397-ODPA InP MSCs. Admittedly, direct use of the terms such as core and periphery without proper characterizations is an overstatement and can be misleading. We have corrected the text in the revised manuscript as shown below to avoid such confusions.

We were not able to find a way to isolate InP MSC-397 from its organic phosphorous-containing by-products. As the result, we were not able to obtain proper elemental analysis. InP MSC-397 was first reported by Cossairt group without any elemental analysis till now, we assume they suffered from the same problem. The Zn content difference is the main composition difference between ODPA-capped InP MSC-397 and Zn-incorporated InP MSC-393.

The purification problem of InP MSC-397 also had inhibited us from obtaining the XRD structural data on InP MSC-397. To respond to the reviewer’s comment, we have performed additional experiments and successfully obtained a sample partially purified but enough to acquire XRD signals. F397-ODPA InP MSCs showed the zinc-blende structure (Figure R13a) as similar to that of F393-InP:Zn MSCs. In addition, both InP MSC-397 and F393-InP:Zn MSC show very similar absorption features from the lowest energy electronic transition to the third transition (Figure R13b). Based on these results, we have expanded the text in the revised manuscript as shown below.

Regarding the issue of comparing the core of F360-InP:Zn MSCs, F408-InP:Zn MSCs, and F393-InP:Zn MSCs, because the core is not a well-characterized entity we have to admit that the comparison is only a part of discussion. We believe the inorganic skeleton of F360-InP:Zn MSCs to be very different from those of F408-InP:Zn MSCs and F393-InP:Zn MSCs. As the result, we believe the core of F360-InP:Zn MSCs to be also very different. We believe F408-InP:Zn MSCs and F393-InP:Zn MSCs share a similar core.

(Revised manuscript page 22, omission of what precedes) Here, compared to F397-ODPA InP MSCs, the blue-shifted absorption spectrum of F393-InP:Zn MSCs delineates the decreased effective

size of ‘quantum-confined core’, which suggests that the incorporated zinc cations may be mostly located at the ‘periphery’ of the MSC. Such Zn incorporations can also reduce the symmetry of F393-InP:Zn MSCs, which yielded the smaller absorption cross section value (Figure 8f) and broad absorption and emission spectral features. (Omission of what follows)

(Revised manuscript page 21, omission of what precedes) Since F393-InP:Zn MSCs have only phosphonate for surface ligands, we prepared F397-ODPA InP MSC, which was previously reported by Gary *et al* as InP MSCs with solely phosphonate surfaces,⁸ and investigated the effect of Zn incorporation on phosphonate-capped InP MSCs. Similar to the case of F393-InP:Zn MSC, XRD returned F397-ODPA InP MSC to be zinc-blende structure (Figure S26). Earlier optical studies compared the pair of 386-MA InP MSC and F399-InP:Cl MSC to pure InP MSC vs. Cl-incorporated InP MSCs. Similarly, the pair of F397-ODPA InP MSC and F393-InP:Zn MSC was chosen here.

Figure R13. (a) XRD pattern of F397-ODPA InP MSCs, (b) UV-vis absorption spectra of F397-ODPA InP MSCs and F393-InP:Zn MSCs.

(Omission of what follows)

Comment 6: What is the composition-structure-property relationship for MA-capped InP MSC-386 (exhibiting one absorption singlet) and ODPA-capped InP MSC-397 (displaying one absorption doublet)?

Our response: We thank the reviewer for the comment. As stated above in response to comment 5, we were not able to obtain elemental analysis data on InP MSC-397. We have obtained XRD patterns for InP MSC-386 and InP MSC-397 (Figure R14). 386-InP MSCs showed the polytwistane structure whereas InP MSC-397 was zinc-blende. We believe the structure has changed from low symmetry

Figure R14. XRD patterns of 386-InP MSCs and F397-InP MSCs.

polytwistane InP MSC-386 to high symmetry zinc-blende InP MSC-397. Such change was reflected in the optical properties.

Steady-state absorption and emission spectra of InP MSC-386 are both rather broad. To the contrary, InP MSC-397 showed the narrow lowest transition at 393 nm with another two electronic transition at 354 nm and 283 nm and strong band-edge emission at 420 nm and tailing over 445 nm.

TA spectra of InP MSC-386 showed the broad and long-lived photoinduced absorption signals in the lower-energy region of band-edge bleaching around 386 nm. The significantly broad and long-lived features are attributable to transitions from the lowest excited electronic state to higher excited electronic states, which is typically observed in molecular systems. To the contrary, InP MSC-397 showed the spectral features typically arising from the exciton relaxation dynamics in quantum-confined systems.

Comment 7: For the InP MSC-386 to Cl-incorporated InP MSC-360 evolution shown in Figure 1c at 80 °C, the authors attributed the absorption peaking at 416 nm as one intermediate species (416-IS). The half width at half-maximum (HWHM) of “416-IS” is 20 nm, the same as that of Cl-incorporated InP MSC-360. Can this intermediate be Cl-incorporated InP MSC-416?

Our response: We thank the reviewer for the insightful comment. We believe the 416-IS to be another Cl-incorporated InP MSC. To better respond to the reviewer’s comment, we have performed additional experiments. The conversion reaction from InP MSC-386 to InP MSC-360 was carried out at a lowered temperature of 25°C. Unfortunately, pure 416-IS was not able to be isolated. However, an intermediate sample which was more rich in the 416-IS could be obtained. The aliquot taken at 10

mins from the 25°C reaction showed the red-shifted absorption because of the higher content of 416-IS than the one previously obtained under conventional 80°C condition (Figure R15a). Aliquots were taken at 10 mins, 6 hrs, and 14 hrs during the conversion reaction from InP MSC-386 to InP MSC-360 at 25°C. Absorption, XRD, and elemental analysis were characterized for those three aliquots along with InP MSC-386 and InP MSC-360 (Figure R15b, 15c, and 15d). The 416-IS contains significant amount of Cl atoms and is considered as another Cl-incorporated InP MSC.

Based on the additional experimental data, the revised manuscript has been also properly expanded as shown below.

(Revised manuscript page 6, omission of what precedes) 416-IS peak was fitted by a Gaussian peak that had 20 nm half width at half-maximum (HWHM) (Figure S2). Though 416-IS could not be purely isolated, a mixture sample rich in 416-IS could be obtained during a conversion from InP MSC-386 to InP MSC-360 at a lowered temperature. Combined analysis based on the absorption, XRD, and elemental analysis suggests 416-IS to be another InP:Cl MSCs (Figure S3). The added InCl_3 *in situ* generates HCl and etches 386-MA InP MSCs. We speculate that these partially-etched

Figure R15. (a) UV-vis absorption spectra of intermediates containing 416-IS synthesized at 80°C or 25°C, (b) UV-vis absorption spectra, (c) XRD patterns and (d) EDS data of aliquots taken during conversion of F360-InP:Cl MSCs from 386-InP MSCs using as-synthesized 386-InP MSCs and 56 equivalents of InCl_3 at room temperature.

386-MA InP MSCs are 416-IS. (Omission of what follows)

Comment 8: For the evolution from InP MSC-386 to Cl-incorporated InP MSC-360 shown in Figure S4b at 50°C and Figure S4c at 25°C, the authors claimed (Lines 141 to 142 on Page 6) “intermediates which have not been properly detected”. Regarding the evolution from InP MSC-386 to Cl-incorporated InP MSC-360 at these different temperatures studied as shown in Figures 1c, S4b, and S4c, any explanation why the experimental observation on “intermediates” is influenced by the temperatures?

Our response: We thank the reviewer for the helpful comment. At 80°C evolution from InP MSC-386 to Cl-incorporated InP MSC-360, an isosbestic point was observed. However, the isosbestic point was missing at the same reaction at 25 or 50°C. Based on the results, we assumed that the concentration of intermediates or the kinds of intermediates should be different by the different reaction temperatures. Different reaction temperature may affect differently for the kinetics of fragmentations and assemblies. Different reaction temperature may also yield different fragment species, partial assemblies of which result in the same final product.

Comment 9: For the Page 9 equation, any numbers can be assigned to x?

Our response: We thank the reviewer for the comment. For an example, in conversion from 386-InP MSCs to F360-InP:Cl MSCs, the x value can be assigned approximately as six by using the elemental analysis data of the reactant and product. However, as stated in the manuscript, we are not arguing that the reaction stated by the equation is the only reaction happening during the conversion.

Comment 10: For the Page 9 equation, if $y = 0$, the reaction becomes an anionic exchange reaction between P and Cl. Any explanation?

Our response: We thank the reviewer for the comment. All the surface atoms are indium, and phosphorous atoms are inside of the inorganic skeleton. As the result, there is no reaction with $y=0$.

Comment 11: All the figures should be improved, including the color usage, the legend font size, and dpi. For example, the Figure 1 legend font size is far too small.

Our response: We thank the reviewer for the comment. We have changed the figures, color usage, the legend font size, and dpi as suggest by the reviewer in Figure 1, Figure 2, Figure 3, Figure 4, Figure 5, Figure 7, Figure 8, and Table of Contents artwork.

Comment 12: The format of the refs should be edited according to the magazine required format.

Our response: We thank the reviewer for the comment. We have corrected the format as suggested.

REVIEWERS' COMMENTS:

Reviewer #1 (Remarks to the Author):

The manuscript appears to have been revised with great care and attention to the reviewer comments. I believe the work now merits publication.

Reviewer #3 (Remarks to the Author):

The authors have tried hard to “answer” the concerns and suggestions raised. Importantly, they need to “improve” their manuscript, based on the concerns and suggestions raised. Although some explanations or interpretations have been provided, the reviewer believes that the authors are able to push harder to do a better job, so that the study will have a bigger impact. In Discussion, the authors should better reflect what is actually known and unknown about these clusters and the implications. Also, the authors need to improve their English.

Before Reviewer 3 Comment 1, “The structural studies do not seem to be convincing.” The authors may address the reason in their revised manuscript why the cluster structures are difficult to identify at present (as partially indicated by The future of colloidal semiconductor magic-size clusters. ACS Nano 2020, 14, 1227–1235).

It is unclear to the reviewer “Our findings highlight the mechanistic insight that MSCs which appear before the growth to QDs transform by multi-step sequential conversions of MSCs or evolutions of MSCs.” Do they want to say “Our studies introduce the possible pathways, regarding the evolution of MSCs which may include multi-step sequential conversions among MSCs”?

Comment 1: the significant figure should be taken care of for the compositions. The authors should address “the composition-structure-property relationship between Cl-incorporated InP MSC-360 and Cl-incorporated InP MSC-399” in their revised version. In Discussion?

Comment 2: the significant figure should be taken care of for the compositions. On Page 12 of the point-by-point response file, the authors seem to write down a lot. The reviewer recommends that the authors should concisely summarize “the composition-structure-property relationship for Zn-incorporated InP MSC-360, Zn-incorporated InP MSC-408, and Zn-incorporated InP MSC-393” in their revised version. In Discussion?

Comment 3: On Page 14 of the response letter, for the response of reviewer 3, comment 3, the authors claimed that “When In(MA)₃ were added into F393-InP : Zn MSCs at room temperature, the absorption peak red-shifted to 404 nm and the second transition disappeared (Figure R9a).”

Here is one example that the authors should carefully improve their English.

Any explanation for the disappearance of the second transition? Is it due to disappearance of the MSCs and the growth of QDs? On Page 15, Figure S8 was presented, showing a sharp absorption peak obtained for a 150 °C sample. Why Figures R8 and R9 are presented, while the edited words on Page 15 mentioned Figures S6 and S7?

Again, for the present Figure R8 on Page 15, the two traces for the 110 and 180 °C samples look similar.

Comment 4: The authors should make a clear statement in their revised version “What is the nature of the evolution? The authors claimed that the two clusters have different compositions”.

Comment 5: The authors should straightforwardly comprehend in their revised version “do they have different cores?”

Comment 6: The authors should provide an answer in their revised version “the composition-structure-property relationship for MA-capped InP MSC-386 (exhibiting one absorption singlet) and ODPA-capped InP MSC-397 (displaying one absorption doublet)?” For example, if the authors do not have an answer, that is fine. The authors may point out such a concern but without an answer, saying “we do not know why MA-capped” This is a kind reminder for to readers to think as well.

Comment 7: The authors edited in their revised manuscript “.....We speculate that these partially-etched 386-MA InP MSCs are 416-IS.” Do the author mean “due to more Cl incorporated”? The reviewer hopes that the authors is able to make their point more directly.

Comment 8: The authors should concisely explain in their revised version “why the experimental observation on “intermediates” is influenced by the temperatures?”

Comment 9: The authors should provide an answer in their revised version “any numbers can be assigned to x”

Comment 10: The authors should improve their revised version answering “ if $y = 0$, ...?”

Comment 11: The authors should improve their figure presentation. For example, 50 °C can be the title of Part a, and does not need to be present for each trace. The four subfigures can be arranged in two panels (two top and two bottom), so do Figure 2.

Figure 8, what is the x axis for Part e? Shall the x axis title of Part f be italic, while the rest titles are not?

There is no need to use “F”, while keep “MSCs”. The author can use, for example, MSC-360 Cl : InP, for their main text and for their figures.

REVIEWERS' COMMENTS:

(Response to reviewer 3)

General Comment 1: The authors have tried hard to “answer” the concerns and suggestions raised. Importantly, they need to “improve” their manuscript, based on the concerns and suggestions raised. Although some explanations or interpretations have been provided, the reviewer believes that the authors are able to push harder to do a better job, so that the study will have a bigger impact. In Discussion, the authors should better reflect what is actually known and unknown about these clusters and the implications. Also, the authors need to improve their English.

Before Reviewer 3 Comment 1, “The structural studies do not seem to be convincing.” The authors may address the reason in their revised manuscript why the cluster structures are difficult to identify at present (as partially indicated by The future of colloidal semiconductor magic-size clusters. *ACS Nano* 2020, 14, 1227–1235).

Our response: We thank the reviewer for the insightful comment. Thanks to the precious comments, we were able to significantly improve the manuscript by the revision. The Discussion section has been properly expanded as shown below.

(Revised manuscript page 24, omission of what precedes) For both InP:Cl and InP:Zn MSCs, evolutions from uni-molecular-like to more QD-like characters were observed. However, structural identifications presented herein for each species were rather limited.¹⁷ It is not simply because the evolutions involve many species but there are limited tools to directly elucidate the structures of MSCs. Characters of each species were extracted and compared by combining data from XRD, NMR, XPS, elemental analysis, and optical studies including TA and Raman. Nevertheless, it is far from pinpointing the structures of each species. Further structural identification is currently on-going for the MSCs using synchrotron radiation, and hopefully will provide better understanding. (omission of what follows)

¹⁷Cristina Palencia, Kui Yu, and Klaus Boldt, *ACS Nano* 2020, 14, 1227–1235.

General Comment 2: It is unclear to the reviewer “Our findings highlight the mechanistic insight that MSCs which appear before the growth to QDs transform by multi-step sequential conversions of MSCs or evolutions of MSCs.” Do they want to say “Our studies introduce the possible pathways, regarding the evolution of MSCs which may include multi-step sequential conversions among MSCs”?

Our response: We thank the reviewer for the insightful comment. As the reviewer nicely pointed out, our studies introduce the possible pathways regarding the evolution of MSCs which may include multi-step sequential conversions among MSCs. Evolutions of MSCs are typically observed during

early stages of nucleation and growth of QDs. It may not be a coincidence that MSCs become more and more QD-like as getting close to QDs. In this vein, we had stated that “Our findings highlight the mechanistic insight that MSCs which appear before the growth to QDs transform by multi-step sequential conversions of MSCs or evolutions of MSCs.”. However, admittedly it was an overstatement. In the revised manuscript, we have revised the discussion as shown below.

(Revised manuscript page 24, omission of what precedes) Our studies introduce multi-step sequential conversions among InP MSCs which evolved to be more QD-like as the conversions proceeded. Evolutions of MSCs are typically observed during early stages of nucleation and growth of QDs. It may not be a coincidence that MSCs become more and more QD-like as getting close to QDs. We are hopeful that further studies lead to mechanistic insight in QD formations and pave the way toward rational design of high-quality QD synthesis and development of new nanostructures that cannot be attained by conventional synthesis. (omission of what follows)

Comment 1: The authors should address “the composition-structure-property relationship between Cl-incorporated InP MSC-360 and Cl-incorporated InP MSC-399” in their revised version. In Discussion?

Our response: We thank the reviewer for the helpful comment. Considering the suggestion, we added discussions about our explanation of two Cl incorporated InP MSCs as follows.

(Revised manuscript page 23, omission of what precedes) 386-InP MSCs and F360-InP:Cl MSCs had similar crystal structures, whereas F399-InP:Cl MSCs showed the transition from polytwistane to a structure similar to zinc-blende InP. F360-InP:Cl MSCs may share a similar inorganic skeleton as 386-InP MSCs because F360-InP:Cl MSCs are from partially ‘etched’ 386-InP MSCs. To the contrary, F399-InP:Cl MSCs have the volume almost two times larger than 386-InP MSCs and F360-InP:Cl MSCs and thus show the more balanced In:P composition, which explains the crystal structure evolution to zinc-blende. (omission of what follows)

Comment 2: On Page 12 of the point-by-point response file, the authors seem to write down a lot. The reviewer recommends that the authors should concisely summarize “the composition-structure-property relationship for Zn-incorporated InP MSC-360, Zn-incorporated InP MSC-408, and Zn-incorporated InP MSC-393” in their revised version. In Discussion?

Our response: We thank the reviewer for the helpful comment. Per the suggestion, we have expanded the discussion in the revised manuscript as shown below.

(Revised manuscript page 23, omission of what precedes) Emission of F360-InP:Zn MSC was very broad and weak, whereas those of F408-InP:Zn MSC and F393-InP:Zn MSC were relatively sharp and strong. F360-InP:Zn MSC had the PL decays that were independent of the emission wavelengths. To the contrary, F408-InP:Zn MSC and F393-InP:Zn MSC showed significant different decays between the PL maximum and tailing regions. F360-InP:Zn MSC showed the emission feature more close to a molecular system whereas F408-InP:Zn MSC and F393-InP:Zn MSC exhibited QD-like band-edge and trap-associated emissions. F408-InP:Zn MSC showed the higher PL QY than F393-InP:Zn MSC and showed the larger amplitude at long decay component when compared to F393-InP:Zn MSC. These PL features indicate that the trap-associated emission is more enhanced in F408-InP:Zn MSC, which accords well with the fact that F408-InP:Zn MSCs contain larger Zn content than F393-InP:Zn MSC. Both F408-InP:Zn MSCs and F393-InP:Zn MSCs exhibited QD-like PL properties, however the characters were slightly different by the population of Zn trap sites. (omission of what follows)

Comment 3: On Page 14 of the response letter, for the response of reviewer 3, comment 3, the authors claimed that “When $\text{In}(\text{MA})_3$ were added into F393-InP : Zn MSCs at room temperature, the absorption peak red-shifted to 404 nm and the second transition disappeared (Figure R9a).” Here is one example that the authors should carefully improve their English. Any explanation for the disappearance of the second transition? Is it due to disappearance of the MSCs and the growth of QDs?

Our response: We thank the reviewer for the helpful suggestion. Admittedly, it is a statement that may cause confusions. We speculate that the disappearance of second transition is due to the change in electronic structure as the doublet feature F393-InP:Zn MSCs converted to a singlet feature 404 nm intermediate. However, the 404 nm intermediate was not fully characterized. We cannot clearly say if the disappearance of the second transition is due to the structural change or anything else such as growth of another species. Further studies are current on-going in this direction, and we hope this issue can be clarified at our forthcoming publication.

Comment 4: On Page 15, Figure S8 was presented, showing a sharp absorption peak obtained for a 150°C sample. Why Figures R8 and R9 are presented, while the edited words on Page 15 mentioned Figures S6 and S7?

Our response: We thank the reviewer for the comment. In the comment “On Page 15, Figure S8 was presented, showing a sharp absorption peak obtained for a 150°C sample.”, we believe the S8 is a typo for Figure R8. In Figure R8 (currently Supplementary Figure S7), the sharp absorption peak obtained for a 150°C sample is from F399-InP:Cl MSCs.

The singlet feature 404 nm intermediate studies are current on-going. In this regard, Figure R9 was

not included in the revised manuscript. Both R8 and R9 show growth of nanoparticles from MSCs though the growth conditions are somewhat different. This manuscript is more to do with the conversions of MSCs and we believe growth of InP nanoparticles using MSCs is beyond the scope of this manuscript. Both Supplementary Figure S6 and Supplementary Figure S7 (also labeled as Fig. R8) demonstrate growth of InP nanoparticles using our MSCs and we believe those two figures are enough for the purpose and justify the omission of Figure R9.

(Revised manuscript page 7, omission of what precedes) Starting from 386-MA InP MSCs at $T_R < 80^\circ\text{C}$, conversion was halted at F360-InP:Cl MSCs and did not further proceed to F399-InP:Cl MSCs (Supplementary Figure 4). At $T_R > 110^\circ\text{C}$ conversion to F399-InP:Cl MSCs occurred and at $T_R > 180^\circ\text{C}$ such conversion was rapidly followed by further growth to NPs (Supplementary Figure 6). Heat-up experiment to 200°C also yielded similar results with zinc-blende a-few-nm InP NPs (Supplementary Figure 7). (Omission of what follows)

Comment 5: Again, for the present Figure R8 on Page 15, the two traces for the 110°C and 180°C samples look similar.

Our response: We thank the reviewer for the comment. Figure R8 on Page 15 show absorption profiles of aliquots taken at different temperatures that include samples at 110°C and at 180°C . However, those two absorption profiles look notably different. Comparison between two traces of the samples at 110°C and 180°C appear in Supplementary Figure 6. In Supplementary Figure 6, the two sample sets look similar because both reactions share the same conversion reaction from 386-MA InP MSCs to F399-InP:Cl MSCs. However, in the case of 180°C samples, growth of InP nanoparticles was additionally observed as indicated by the development of absorption tail to the long wavelengths.

Comment 6: The authors should make a clear statement in their revised version “What is the nature of the evolution? The authors claimed that the two clusters have different compositions”.

Our response: We thank the reviewer for the suggestion. Per the suggestion, we have expanded the text in the revised manuscript with an additional figure (what used to be Figure R12) as shown below.

(Revised manuscript page 20, omission of what precedes) In conversion from F408-InP:Zn MSCs to F393-InP:Zn MSCs, absorption and FT-IR spectra were followed (Supplementary Figure 25). Adding larger amount of ODPA significantly accelerate the conversion to F393-InP:Zn MSCs. IR spectra showed that the binding mode of phosphonate ligands changed more to divalent as the conversion proceeded. Narrowing of the P-O(H) vibration peaks between 1000 and 1200 cm^{-1} was observed, which attributes to the change to deprotonated ODPA species from mixed ODPA species of (partially) protonated and deprotonated ODPA. The deprotonated ODPA ligands should bind to the surface In

atoms more strongly than protonated ODPA, and this should make the final F393-InP:Zn MSCs to be very stable. Thermodynamically, the nature of the evolution can be considered as the enthalpy gain by exchanging the initial ligands to a stronger deprotonated ODPA ligand. (Omission of what follows)

Supplementary Figure 25. (a) UV-vis absorption spectra and (b) FT-IR spectra of aliquots taken during direct synthesis of F393-InP:Zn MSCs from $\text{In}(\text{Ac})_3$, ODPA, $(\text{TMS})_3\text{P}$ and $\text{Zn}(\text{SA})_2$ precursors at 300°C . The molar ratio of $\text{In}(\text{Ac})_3$ to ODPA is 1:0.925. (c) UV-vis absorption spectra and (d) FT-IR spectra of aliquots taken during direct synthesis of F393-InP:Zn MSCs from $\text{In}(\text{Ac})_3$, ODPA, $(\text{TMS})_3\text{P}$ and $\text{Zn}(\text{SA})_2$ precursors at 300°C . The molar ratio of stearate to ODPA is 1:0.875.

Comment 7: The authors should straightforwardly comprehend in their revised version “do they have different cores?”

Our response: We thank the reviewer for the comment. We believe that F408-InP:Zn MSCs and F393-InP:Zn MSCs share a similar inorganic skeleton close to zinc-blende core but differ in Zn trap site density and that F393-InP:Zn MSCs and F397-ODPA InP MSCs may share a more similar core (both zinc-blende and electronically doublet absorption feature) but differ in Zn trap site existence. We believe the initial F360-InP:Zn MSCs to have the inorganic skeleton disparate from the rest of the series. We have properly added the discussion in the revised manuscript as shown below.

(Revised manuscript page 22, omission of what precedes) Here, compared to F397-ODPA InP MSCs, the blue-shifted absorption spectrum of F393-InP:Zn MSCs may delineate the decreased effective size of ‘quantum-confined core’, which suggests that the incorporated Zn cations may be mostly located at

the ‘periphery’ of the MSC. Such Zn incorporations can also reduce the symmetry of F393-InP:Zn MSCs, which yielded the smaller absorption cross section value (Figure 9f) and broad absorption and emission spectral features. We believe that F408-InP:Zn MSCs and F393-InP:Zn MSCs share a similar inorganic skeleton close to zinc-blende core but differ in Zn trap site density and that F393-InP:Zn MSCs and F397-ODPA InP MSCs may share a more similar core (both zinc-blende and electronically doublet absorption feature) but differ in Zn trap site existence. We believe the initial F360-InP:Zn MSCs to have the inorganic skeleton disparate from the rest of the InP:Zn MSCs series. (omission of what follows)

Comment 8: The authors should provide an answer in their revised version “the composition-structure-property relationship for MA-capped InP MSC-386 (exhibiting one absorption singlet) and ODPA-capped InP MSC-397 (displaying one absorption doublet)?” For example, if the authors do not have an answer, that is fine. The authors may point out such a concern but without an answer, saying “we do not know why MA-capped” This is a kind reminder for to readers to think as well.

Our response: We thank the reviewer for the comment. For structure-optical property relationship for 386-MA InP MSCs and F397-ODPA InP MSCs, 386-MA InP MSC showed the polytwistane structure and broad absorption feature at 386 nm whereas F397-ODPA InP MSC was zinc-blende with narrow absorption feature at 397 nm. We believe the structure has changed from low symmetry polytwistane 386-MA InP MSC to high symmetry zinc-blende F397-ODPA InP MSC. Such change was reflected in the optical properties. However, we have not fully proved the composition-structure-optical property relationship between 386-MA InP MSC and F397-ODPA InP MSC directly. For the detailed discussion, the revised manuscript has been expanded as shown below.

(Revised manuscript page 22, omission of what precedes) To compare structure-optical property relationship between 386-MA InP MSCs and F397-ODPA InP MSCs, 386-MA InP MSC showed the polytwistane structure and broad absorption feature at 386 nm whereas F397-ODPA InP MSC was zinc-blende with narrow absorption feature at 397 nm. TA spectra of 386-MA InP MSC showed the broad and long-lived photoinduced absorption signals in the lower-energy region, which is attributable to transitions in molecular systems. To the contrary, F397-ODPA InP MSC showed the spectral features typically arising from the exciton relaxation dynamics in quantum-confined systems. (omission of what follows)

Comment 9: The authors edited in their revised manuscript “.....We speculate that these partially-etched 386-MA InP MSCs are 416-IS.” Do the author mean “due to more Cl incorporated”? The reviewer hopes that the authors is able to make their point more directly.

Our response: We thank the reviewer for the helpful suggestion. We believe that these partially

etched 386-MA InP MSCs are 416-IS not simply because of more Cl incorporation. 386-MA InP MSCs and 416-IS share a similar crystal structure as confirmed by XRD. Composition analysis returned more In-rich for 416-IS, the phosphorus contents decreased from In:P:Cl of 1:0.54:0 to 1:0.39:0.23. It suggests two possible pathways to 416-IS from 386-InP MSCs: (1) addition of indium and chlorine to 386-InP MSCs and (2) removal of phosphorus by etching and addition of chloride. NMR observation of PH_3 evolution in conversion from 386-MA InP MSCs to 416-IS strongly supports the latter for the possible pathway. The revised manuscript has been altered to more directly point the discussion.

(Revised manuscript page 5, omission of what precedes) 386-InP MSCs were converted into F360-InP:Cl MSCs via a 416 nm intermediate species (416-IS) that showed the LEET at 416 nm. 416-IS peak was fitted by a Gaussian peak that has 20 nm half width at half-maximum (HWHM) (Supplementary Figure 2). Though 416-IS could not be purely isolated, a mixture sample rich in 416-IS could be obtained during a conversion from 386-InP MSC to F360-InP:Cl MSC at a lowered temperature. Combined analysis based on the absorption, XRD, and elemental analysis suggests 416-IS to be another InP:Cl MSCs (Supplementary Figure 3). The added InCl_3 in situ generates HCl and etches 386-MA InP MSCs as evolving PH_3 (vide infra). Composition analysis showed more In-rich for 416-IS than 386-MA InP MSC. We speculate that 416-IS has the inorganic skeleton that is from partially-etched 386-MA InP MSC. (Omission of what follows)

Comment 10: The authors should concisely explain in their revised version “why the experimental observation on “intermediates” is influenced by the temperatures?”

Our response: We thank the reviewer for the helpful suggestion. Per the suggestion, we have added concise explanation on why the experimental observation on observation on “intermediates” is influenced by the temperatures in the revised manuscript as shown below with an additional reference.

(Revised manuscript page 6, omission of what precedes) Different T_R may have affected differently for the kinetics of fragmentations and their assemblies.²⁸ Different T_R may have also yielded different fragments and partial assemblies of which resulted in the same final product. Alternatively, 416-IS may have gone through some sort of amorphization before crystallize to F360-InP:Cl MSCs. (Omission of what follows)

²⁸ Max R. Friedfeld, Dane A. Johnson, and Brandi M. Cossairt, *Inorg. Chem.* **2019**, *58*, 803–810.

Comment 11: The authors should provide an answer in their revised version “any numbers can be assigned to x”

Our response: We thank the reviewer for the comment. For an example, in conversion from 386-InP MSCs to F360-InP:Cl MSCs, the x value can be assigned approximately as six by using the elemental analysis data of the reactant and product. As suggested, the revised manuscript has been also properly expanded as shown below.

(Revised manuscript page 9, omission of what precedes)

Using the elemental analysis data of the reactant and product (Table 1), the x value herein can be assigned approximately as six. (Omission of what follows)

Comment 12: The authors should improve their revised version answering “if y = 0, ...?”

Our response: We thank the reviewer for the comment. As suggested, the revised manuscript has been properly expanded with the case y=0 as shown below.

(Revised manuscript page 9, omission of what precedes) However, the reactions stated above do not entirely describe the conversion reaction from 386-InP MSCs to F360-InP:Cl MSCs. When y = 0, the reaction describes an anionic exchange reaction between P and Cl. However, such case is unlikely because all the surface atoms are In for 386-InP MSCs. (Omission of what follows)

Comment 13: The authors should improve their figure presentation. For example, 50 °C can be the title of Part a, and does not need to be present for each trace. The four subfigures can be arranged in two panels (two top and two bottom), so do Figure 2.

Our response: We thank the reviewer for the comment. Figure 1 and Figure 2 were improved in the revised manuscript and we have also corrected the x axis and style in Figure 8.

Figure 2. Syntheses of F360-InP:Cl MSCs and F399-InP:Cl MSCs. UV-vis absorption spectra of aliquots taken over time during (a) direct synthesis of F360-InP:Cl MSCs from indium acetate, MA, InCl_3 and $(\text{TMS})_3\text{P}$ at 50°C . The molar ratio of indium acetate, MA, to InCl_3 is 1:3:0.6. (b) Direct synthesis of F399-InP:Cl MSCs from $\text{In}(\text{Ac})_3$, InCl_3 , HMA and $(\text{TMS})_3\text{P}$ at 80°C . The molar ratio of $\text{In}(\text{Ac})_3$, HMA to InCl_3 is 1:3:1. (c) Conversion synthesis of F360-InP:Cl MSCs from as-synthesized 386-MA InP MSCs using 56 equivalents of InCl_3 at 80°C . (Red arrow: seemingly isosbestic point) (d) Conversion synthesis of F399-InP:Cl MSCs from F360-InP:Cl MSCs by heating.

Figure 3. Conversion from 386-InP MSC to F360-InP:Cl MSCs. UV-vis absorption spectra of aliquots taken over time (a) during conversion from as-synthesized 386-OA InP MSCs to F360 InP:Cl MSCs using 56 equivalents of InCl_3 , (b) during conversion reaction from purified 386-OA InP MSCs using 56 equivalents of InCl_3 , and (c) during conversion from purified 386-OA InP MSCs to F360 InP:Cl MSCs using 56 equivalents of InCl_3 and 112 equivalents of OA. (d) $^{31}\text{P}\{^1\text{H}\}$ NMR spectra of 386-MA InP MSCs and the converted F360-InP:Cl MSCs.

Figure 8. Optical characterizations of F393-InP:Zn MSCs and F397-InP MSCs. Steady-state absorption and PL spectra of (a) F397-ODPA InP MSCs and (b) F393-InP:Zn MSCs. TA spectra of (c) F397-ODPA InP MSCs and (d) F393-InP:Zn MSCs. (e) Normalized TA spectra at 2 ps of F397-ODPA InP MSCs and F393-InP:Zn MSCs. (Inset: normalized TA decay profile of F397-ODPA InP MSCs at 397 and of F393-InP:Zn MSCs at 393 nm.) (f) Plot for normalized LEET bleach signals for F397-ODPA InP MSCs and F393-InP:Zn MSCs with the estimated absorption cross section values σ_{abs} (cm^{-1}).

Comment 14: There is no need to use “F”, while keep “MSCs”. The author can use, for example, MSC-360 Cl:InP, for their main text and for their figures.

Our response: We thank the reviewer for the comment. Indeed, there is no need to use “F” in our nomenclature. However, we believe the definition of MSC does not yet have a consensus in the scientific community. We intended to use “F” to clearly define our samples as families of plural isomers. For an example, we intentionally did not use “F” for 386-InP MSC because it is a single species. This may appear odd but we wish to express the terms with such rigorousness.